# Reconstructing five decades of sediment export from two glaciated high-alpine catchments in Tyrol, Austria, using nonparametric regression

Lena Katharina Schmidt[1], Till Francke[1], Peter Martin Grosse[1], Christoph Mayer[2], Axel Bronstert[1]

[1]Institute of Environmental Sciences and Geography, University of Potsdam, Potsdam, 14476, Germany
[2]Bavarian Academy of Sciences and Humanities, Munich, 80539, Germany

*Correspondence to*: Lena Katharina Schmidt (leschmid@uni-potsdam.de)

**Abstract.** Knowledge on the response of sediment export to recent climate change in glaciated areas in the European Alps is limited, primarily because long-term records of suspended sediment concentrations (SSC) are scarce. Here we tested the estimation of sediment export of the past five decades using Quantile Regression Forest (QRF), a non-parametric, multivariate regression based on Random Forests. The regression builds on short-term records of SSC and long records of the most important hydro-climatic drivers (discharge, precipitation and air temperature (QPT)). We trained independent models for two nested and partially glaciated catchments, Vent (98 km²) and Vernagt (11.4 km²), in the Upper Ötztal in Tyrol, Austria (1891 to 3772 m a.s.l.), where available QPT records start in 1967 and 1975. To assess temporal extrapolation ability, we used two 2-year SSC datasets at gauge Vernagt, which are almost 20 years apart for a validation. For Vent, we performed a five-fold cross-validation on the 15 years of SSC measurements. Further, we quantified the number days where predictors exceeded the range represented in the training dataset, as the inability to extrapolate beyond this range is a known limitation of QRF. Finally, we compared QRF performance to sediment rating curves (SRC). We analysed the modelled sediment export time series, the predictors and glacier mass balance data for trends (Mann-Kendall test and Sen's slope estimator) and step-like changes (using the widely applied Pettitt's test and a complementary Bayesian approach).

Our validation at gauge Vernagt demonstrated that QRF performs well in estimating past daily sediment export (Nash-Sutcliffe efficiency (NSE) of 0.73) and satisfactory for SSC (NSE of 0.51), despite the small training dataset. Temporal extrapolation ability of QRF was superior to SRC, especially in periods that contained high SSC events, which demonstrated the ability of QRF to model threshold effects. Days with high SSC tended to be underestimated, but the effect on annual yields was small. Days with predictor exceedances were rare, indicating a good representativity of the training dataset. Finally, the QRF reconstruction models outperformed SRC by about 20 percent-points of explained variance. Significant positive trends in the reconstructed annual suspended sediment yields were found at both gauges, with distinct step-like increases around 1981. This was linked to increased glacier melt, which became apparent through step-like increases in discharge at both gauges as well as change points in mass balances of the two largest glaciers in the Vent catchment. We identified exceptionally high July temperatures in 1982 and 1983 as a likely cause. In contrast, we did not find coinciding change points in precipitation. Opposing trends at the two gauges after 1981 suggest different timings of 'peak sediment'. We conclude that, given large enough training datasets, the presented QRF approach is a promising tool with the ability to deepen our understanding of the response of high-alpine areas to decadal climate change.

## 1    Introduction

Sediment production rates per unit area are highest in the world's mountains (Hallet et al., 1996), headed by modern glaciated basins (Hinderer et al., 2013). As a consequence, sediments transported from these rapidly deglaciating high alpine areas have substantial effects on downstream hydropower production and reservoir sedimentation (Schöber and Hofer, 2018; Guillén-Ludeña et al., 2018; Li et al., 2022), flood hazard (Nones, 2019; Brooke et al., 2022) as well as water quality, nutrient and contaminant transport and aquatic habitats (Gabbud and Lane, 2016; Bilotta and Brazier, 2008; Vercruysse et al., 2017) and impact global sediment and biochemical balances (Herman et al., 2021). Thus, there is considerable interest in water resource research and management to gain better understanding of sediment dynamics in high alpine areas, also to mitigate and adapt to future changes. However, there is still limited quantitative understanding of sediment transport in high-alpine rivers and their relation to changes in climatic forcing over temporal scales relevant to investigating changes associated with anthropogenic climate change, i.e. at decadal and centennial scales as opposed to longer ones (Huss et al., 2017; Antoniazza and Lane, 2021; Herman et al., 2021). This is partly owed to the complexity of sediment dynamics in high-alpine areas, which are the result of an intricate system of climatic forcing and hydro-geomorphological processes (Costa et al., 2018; Vercruysse et al., 2017; Zhang et al., 2022).

A significant body of knowledge exists on how some of the components of these complex systems have changed in recent decades due to rising temperatures. Cryospheric changes include widespread and accelerating glacier retreat (Abermann et al., 2009; Sommer et al., 2020) and reduced extent and duration of snow cover (Beniston et al., 2018; Chiarle et al., 2021). As a result, hydrological regimes are changing from glacial to nival and from nival to pluvial regimes (Beniston et al., 2018), which results in changes in water quantities (Vormoor et al., 2015; Wijngaard et al., 2016), streamflow variability (van Tiel et al., 2019) and hydrograph timing (Kormann et al., 2016; Kuhn et al., 2016; Hanus et al., 2021; Rottler et al., 2020, 2021). These hydrological changes can translate to changes in erosivity, sediment transport capacities and fluvial erosion. At the same time, sediment supply changes, as glacier retreat uncovers large amounts of sediment previously inaccessible to pluvial and fluvial erosion (Carrivick and Heckmann, 2017; Leggat et al., 2015), subglacial sediment discharge transiently increases (Costa et al., 2018; Delaney and Adhikari, 2020) and continuing permafrost thaw destabilizes slopes and facilitates mass movements (Huggel et al., 2010, 2012; Beniston et al., 2018; Savi et al., 2020). Adding to this, erosive precipitation has a higher chance of affecting unfrozen material during prolonged snow-free periods (Kormann et al., 2016; Rottler et al., 2021; Wijngaard et al., 2016).

Yet the magnitude of these impacts is catchment-specific, as it depends e.g. on the area occupied by glaciers and permafrost or basin hypsometry (Huss et al., 2017) and is thus not easily transferable from one site to another. Aggravatingly, high-alpine sediment dynamics are highly variable over time, so that long time series are required to assess systematic changes. Yet, most records are too short for such analyses and long-term data are extremely rare, especially in glaciated headwaters, which are often especially challenging to monitor.

To our knowledge, only very few examples of decadal sediment records from the European Alps exist in the current literature, as opposed to their availability for e.g. High-mountain Asia (Singh et al., 2020; Li et al., 2020, 2021; Zhang et al., 2021), the Andes (Vergara et al., 2022) or the Arctic (Bendixen et al., 2017a) (for an extensive review, see Zhang et al., 2022). Costa et al. (2018b) report on an exceptional record of suspended sediment concentrations from the Upper Rhône basin, Switzerland, of almost five decades, albeit these recordings are severely affected by anthropogenic impacts (hydropower generation and gravel mining) and integrate over an area of 5340 km². Michelletti and Lane (2016) and Lane et al. (2017) reconstructed coarse sediment export from

hydropower intake flushings at decadal scales in three small catchments in the Hérens Valley, Switzerland, however, not taking into account the amount of suspended sediment transport, which is often at least as large as the amount transported as bedload (Schöber and Hofer, 2018; Mao et al., 2019; Turowski et al., 2010). Further long-term sediment records can be inferred from sediment stratigraphy (e.g., Bogen, 2008; Lane et al., 2019) – yet such studies are of course confined to catchments where lakes or reservoirs are present. To compensate for this lack of measurement data, we aim to estimate longer-term past suspended sediment dynamics based on the available shorter records of suspended sediment concentrations.

Quantile Regression Forest (QRF) (Meinshausen, 2006) represents an approach that has been successfully applied to modelling suspended sediment concentrations in past geomorphological studies, in badland-dominated catchments in Spain (Francke et al., 2008a, b) and in a tropical forest in Panama (Zimmermann et al., 2012). QRF is a multivariate, non-parametric regression technique based on Random Forests, from the category of machine learning (ML) approaches, which generally seek to identify patterns from complex data (Tahmasebi et al., 2020). Comparative studies have reported that QRF performs favorably compared to sediment rating curves and generalized linear models (Francke et al., 2008a) and that performance of Random Forest (which QRF are based on) was superior to support-vector machines and artificial neural networks (Al-Mukhtar, 2019) in modelling suspended sediment concentrations. As an advantage to other ML approaches, QRF allow to quantify the model uncertainty through estimating prediction accuracy (Francke et al., 2008a; Al-Mukhtar, 2019).

Thus, the first objective of this study was to extensively test QRF as an approach for estimating past suspended sediment dynamics at decadal scales in high alpine areas. Previous studies have included proxies for drivers of sediment transport and erosive processes, e.g. precipitation, discharge, seasonality and antecedent conditions (Francke et al., 2008a, b; Zimmermann et al., 2012). We built on this setup and adapted it by including air temperature as a predictor, since many processes relevant to sediment dynamics in high alpine areas are temperature-sensitive (e.g. snow- and glacier melt, thawing of topsoil, etc.). To assess model performance, we evaluated several validations and compared the results to sediment rating curves based on data from the two gauges "Vent Rofenache" and "Vernagt" that are located in a nested catchment setup in the Rofental, within the Upper Ötztal in Tyrol Austria. This nested setup provides a favorable opportunity to test the QRF model and gives a good overview of sediment dynamics in this catchment.

The second objective of this study is to examine the resulting estimates of annual suspended sediment yields with respect to changes at decadal scales. Thus, we analyzed the time series for trends, some of which could be expected e.g. due to ongoing temperature increase. However, the possibility of sudden, tipping-point-like shifts in response to climatic changes has been suggested for cryospheric geomorphic systems as well (Huggel et al., 2012) and especially sediment dynamics (Vercruysse et al., 2017) and indeed step-like increases in suspended sediment concentrations have been observed in other catchments (Costa et al., 2018; Li et al., 2020, 2021; Zhang et al., 2021). Hence, we used change point detection methods to assess whether the detected trends are gradual or follow a step-like pattern. We extended these analyses to the predictors (temperature, discharge and precipitation) as well as annual mass balances to assess possible reasons for changes in suspended sediment yields.

## 2   Study area

The study area is located in the "Rofental", a valley in the Upper Ötztal in the Tyrolean Alps, Austria. For more than 100 years, the area has been subject to intense glaciological and hydrometeorological research, yielding outstanding data sets (Strasser et al., 2018). The entire catchment is 98.1 km² upstream gauge "Vent Rofenache"

(hereafter "Vent") and nested within is the 11.4 km² catchment upstream gauge "Vernagt" (see Fig. 1). The gauge Vent is run by the Hydrographic Service of Tyrol; and gauge Vernagt by the Bavarian Academy of Sciences and Humanities. The catchments' elevations range from 1891 m a.s.l. at gauge Vent and 2635 m at gauge Vernagt to the summit of Wildspitze, the highest peak in Tyrol, at 3772 m.

The study area's shielded location in the inner Alpine region leads to the relatively warm and dry climate (Kuhn et al., 1982). Mean annual temperature and precipitation at the gauge in Vent are 2.5°C and about 660 mm (Hanzer et al., 2018; Hydrographic yearbook of Austria, 2016; Strasser et al., 2018), but a considerable precipitation gradient with elevation of about 5 % per 100 m has been described (Schöber et al., 2014).

Both catchments are heavily glaciated, with approximately 28 % and 64 % glacier cover in 2015 (Buckel and Otto, 2018). The two largest glaciers within the Vent catchment, Vernagtferner and Hintereisferner have been systematically observed since the 1950s and 60s and have shown accelerating retreat since the beginning of the 1980s (Abermann et al., 2009), which is expected to continue in the future (Hanzer et al., 2018).

The catchment is drained by the river Rofenache, which flows into the Ötztaler Ache, one of the largest tributaries to the river Inn. The hydrological regime (glacial to nival) shows a pronounced seasonality as most of the discharge is fed by snow and glacier melt and about 89 % of the discharge in Vent occurs from April to September (Schmidt et al., 2022b; Hanzer et al., 2018).

The Ötztal Alps are part of the Ötztal-Stubai massif within the crystalline central Eastern Alps and the catchment geology is dominated by biotite-plagioclase, biotite and muscovite gneisses, variable mica schists and gneissic schists (Strasser et al., 2018). The land cover of higher elevations is dominated by glaciers, bare rock or sparsely vegetated terrain whereas mountain pastures and coniferous forests are present at lower altitudes.

Suspended sediment concentrations at the gauge in Vent showed to be the highest in an Austria-wide comparison (Lalk et al., 2014) and annual suspended sediment yields are about 1500 t/km² on average (Schmidt et al., 2022b). Suspended sediment dynamics showed an even more pronounced seasonality compared to discharge, as 99 % of the annual suspended sediment yields are transported from April to September (ibid.).

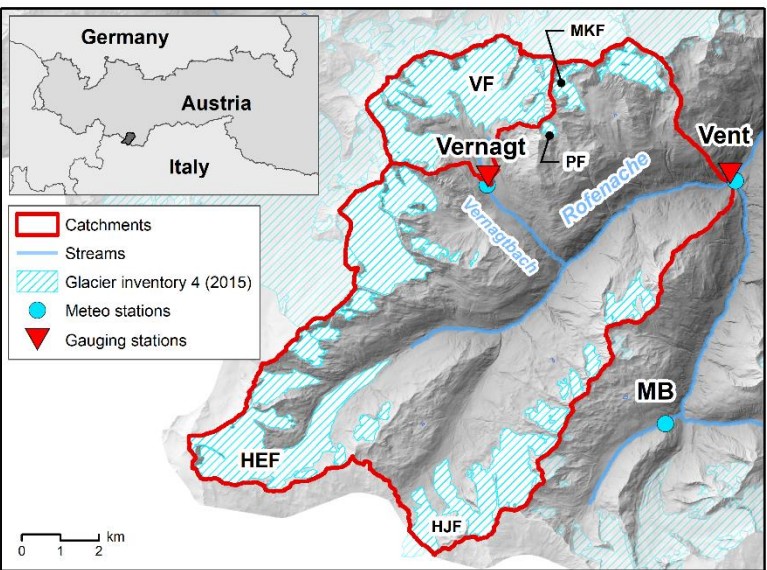

**Figure 1 Map of the catchment area upstream gauge Vent, and nested within the Vernagt catchment, with glaciers Vernagtferner (VF) and Hintereisferner (HEF) as well as the measurement station Martin-Busch-Hütte (MB). Smaller glaciers Hochjochferner, Platteiferner and Mitterkarferner are denoted by HJF, PF and MKF, respectively. Sources: 10 m DTM of Tyrol** *(Land Tirol, 2016)***, glacier inventory 4 (2015)** *(Buckel and Otto, 2018)***, rivers from tiris open government data** *(Land Tirol, 2021)***.**

## 3 Methods

In essence, we trained a non-parametric model on suspended sediment concentrations and predictors (discharge, temperature, precipitation) and then used this model and long-term records of the predictors to reconstruct past suspended sediment concentrations (SSC) and derive annual suspended sediment yields. We then analyzed the resulting time series with respect to trends and change points.

In the following section, we briefly describe the general modeling approach and the validations, the Quantile Regression Forest (QRF) approach and the employed predictors, the input data and their preparation for the model, as well as the statistical tools used for the analysis of the results.

In our analyses, we focus on (annual) suspended sediment yields instead of concentrations, due to the very strong seasonality in discharge (Schmidt et al., 2022b). This gives more weight to days with higher discharge, which are more influential for overall sediment export to downstream areas and the potential resulting problems, as opposed to e.g. mean annual SSC, which weigh days at the beginning and end of the season, with low discharge and low concentrations, and days during the glacier melt period equally. We use the term 'reconstructing' to describe the estimates of SSC (or the resulting yields) from our model simulations.

### 3.1. General modeling approach and adaptations to conditions at the two gauges

We combined the analysis of the two gauges Vent and Vernagt to gain a more reliable and comprehensive understanding of past sediment dynamics in the Rofental. In this, the data situations at the two gauges bear different challenges and opportunities.

At gauge Vent, continuous turbidity-derived SSC time series have been recorded in high temporal resolution (15 minutes) since 2006, providing abundant training data for our model. Additionally, the long-term predictor data are available back until 1967, facilitating insights into long-term changes in catchment dynamics - yet only in daily resolution, which predetermines the temporal resolution of the reconstruction model. This is challenging as sediment concentrations vary considerably during one day, leaving us with the need to assess whether a daily model adequately represents sediment dynamics.

At gauge Vernagt, the availability of hourly discharge (Q), precipitation (P) and temperature (T) data back until 1974 is remarkable. Yet, turbidity-derived SSC data have only been recorded for the years 2000, 2001, 2019 and 2020 (see Fig. 2, upper panel "Vernagt", left side, plots labelled "SSC"). Additionally, the data in 2019 and 2020 are affected by episodic siltation and periods when the turbidity sensor reached saturation: 0.47 % of the two summers of data were affected by saturation and about 8 % were affected by siltation. The latter was mainly due to one period of about 16 days in August 2019. Additionally, there were three shorter incidents (1.5 hours to 1.5 days), two in 2019 and one in 2020. These issues first need to be dealt with in order to provide accurate training data for our model, as it is sensitive to the range of values represented in the training data (Francke et al., 2008a). On the other hand, the 16-year gap between the measurement periods provides the rare opportunity to verify the model's skill in estimating past SSC against real measurement data.

To address these issues and benefit from these opportunities, we preformed three preparatory steps before the final reconstruction models (Fig. 2):

1. **Gap-filling model:** At gauge Vernagt, we trained a model on SSC determined from 131 water samples and the predictors (Q, P, T) in the highest possible (i.e. 10-minute) resolution (Fig. 2, upper panel). We used the resulting modelled SSC to replace periods in the 2019/20 SSC data that were affected by siltation or sensor saturation. As the sampling scheme was customized to cover hydro-sedimentological conditions as widely as possible (see Schmidt et al., 2022), the water samples also partially cover the periods to be addressed by the gap filling.

2. **Validation A - temporal resolution:** we ran models in both hourly and daily resolution on all available training data at gauge Vernagt to assess the error magnitude due to the coarser temporal resolution in the final reconstruction models.

3. **Validation B – extrapolation:** we trained a model on the corrected Vernagt SSC data of 2019 and 2020 in daily resolution and estimated SSC back until 2000 for validation against the 2000-2001 measurement data (Fig. 2, center).

Finally, we ran the 'reconstruction models' in daily resolution (for consistency between the two gauges), taking into account all available training data to estimate SSC back until 1967 and 1975, respectively.

Beyond Validation A and B, we assessed whether and how often the ranges of the predictors in the training data were exceeded during the reconstruction period, since it is a known limitation of QRF that it is not possible to extrapolate beyond the range of values represented in the training data (Francke et al., 2008a). For example, if the discharge measured on a day in the reconstruction period exceeds the maximum discharge within the training period ($Q_{train,\ max}$), the model potentially underestimates SSC on that day.

Additionally, we compared QRF performance to performance of sediment rating curves, one of the most commonly used approaches to estimate suspended sediment concentrations (Vercruysse et al., 2017), by fitting a power function between SSC and Q of the form $SSC = a \cdot Q^b$, where a and b are regression coefficients.

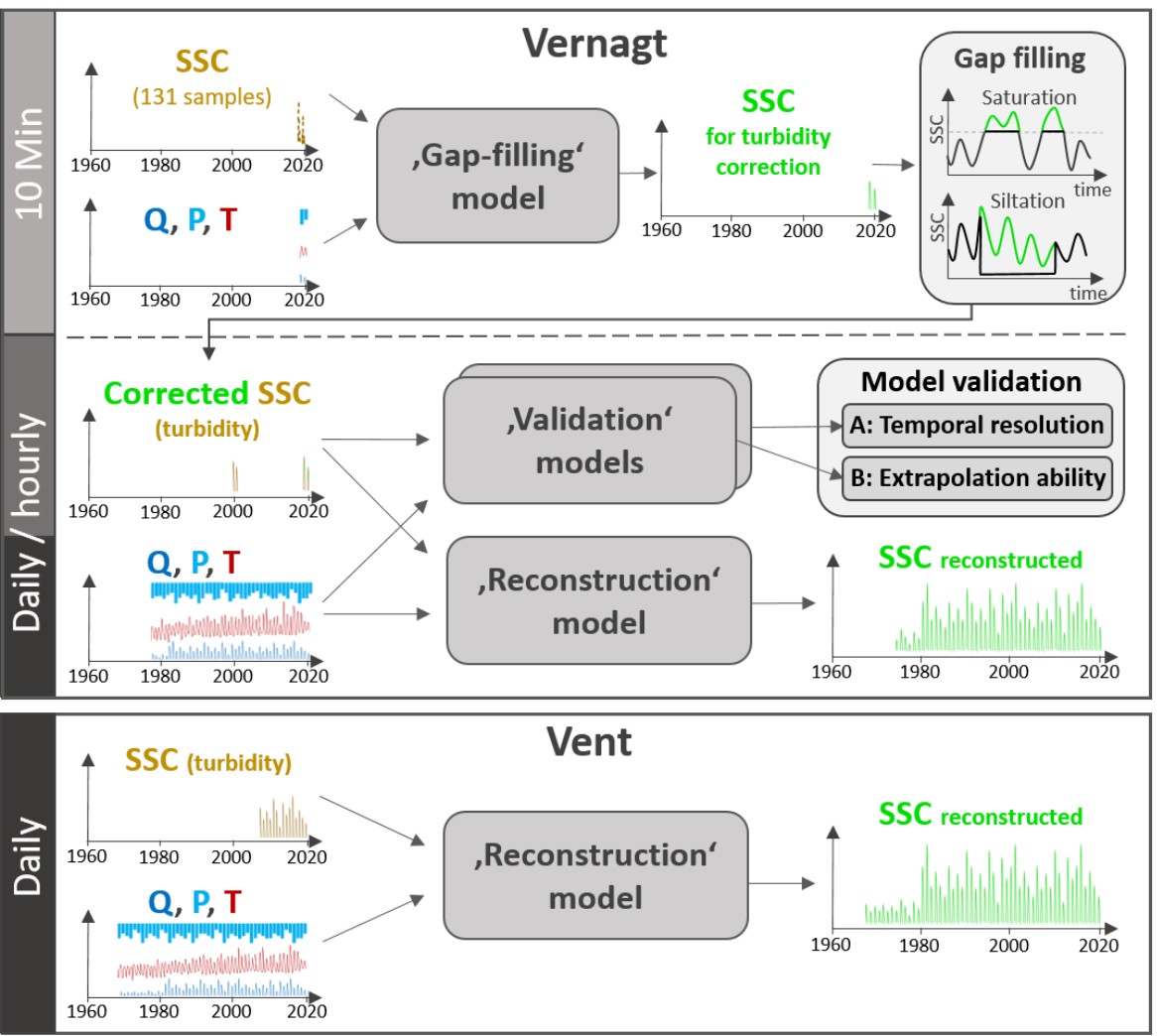

**Figure 2: Overview of modelling approach. SSC = suspended sediment concentration; Q = discharge; P = precipitation; T = temperature.**

### 3.2. Quantile Regression Forest (QRF) for modelling suspended sediment concentrations

To reproduce suspended sediment dynamics, the desired models have to account for a multitude of processes controlling SSC (discharge dynamics, temperature-dependent activation of sediment sources and transport processes, precipitation, antecedent wetness conditions, etc.), need to deal with non-normal distributions of both predictor and response variables and have to handle correlations between the predictors (Zimmermann et al., 2012).

Quantile Regression Forests (QRF) (Meinshausen, 2006) represent a multivariate approach that can deal with non-
225 linearity, interactions and non-additive behavior without making assumptions on underlying distributions, and performed favorably in reproducing sediment dynamics as compared to generalized linear models or sediment rating curves (Francke et al., 2008a). QRF are a generalization of Random Forests (RF) regression tree ensembles (Breiman, 2001). Regression trees (a.k.a. CARTs) (Breiman et al., 1984) apply recursive rule-based data partitioning in order to group data with similar values for the response variable (Francke et al., 2008a). To produce
a "forest", RF and QRF employ an ensemble of these trees, each one grown on a random subset (bootstrap sample) of the training data. Predictive performance is evaluated on those parts of the data that are not considered in the bootstrap sample, i.e. the out-of-bag data (OOB) (ibid.). Model predictions are then obtained from the mean of

predictions of each single tree (RF) or based on the distribution of these single-tree predictions (QRF). As QRF keeps the value of all observations within a node, it enables the quantification of the model uncertainty with the help of these inherent ensemble characteristics. This represents an advantage to other non-parametric approaches applied to SSC modelling, such as traditional fuzzy logic or artificial neural networks (ibid.).

Building on the model setup developed by (Francke et al., 2008b; Francke, 2017; Zimmermann et al., 2012), we chose the predictors to represent proxies for processes important to sediment dynamics in high-alpine areas. For example, discharge is crucial for sediment transfer and, potentially, channel erosion, while precipitation drives hillslope erosion and mass wasting events. We extended the set of primary predictors by adding air temperature, as many processes determining sediment dynamics in high alpine are temperature-sensitive (e.g. the activation of sediment sources, such as the occurrence of rain vs. snow, the availability of sub- and proglacial sediments and their transport through glacier meltwaters, but also potential permafrost thaw). From these three primary predictors (discharge, precipitation and temperature, QPT), we derived information on antecedent conditions for each time step by computing ancillary predictors derived thereof. These ancillary predictors can capture that, e.g., longer-term discharge behavior is linked to potential exhaustion of sediment sources or storage, prolonged warm periods may lead to increased glacier ablation associated with intensified transport of glacial sediment, and high antecedent moisture conditions prior to a precipitation event may favor mass movements. To keep correlation between these derived predictors as low as possible, we used non-overlapping windows of increasing sizes (e.g. 24 hours, 24 – 72 hours, 72 hours to 7 days and 7 days to 20 days ahead of each time step) to compute sums of the primary predictors (Zimmermann et al., 2012). To complete the set of predictors, we used the day of year to capture the pronounced seasonality in high alpine sediment dynamics (Schmidt et al., 2022b), as well as the rate of change in discharge (see also Francke et al., 2008b; Zimmermann et al., 2012).

Furthermore, we tuned the model with respect to the length of the antecedent periods considered: We optimized the length of the time windows for the highest model performance with respect to daily / hourly SSC using the Nash-Sutcliffe efficiency index (see section 3.2.1). Besides the length of the antecedent periods, the two most important hyper-parameters for the QRF are the number of trees in a "forest" and the number of selected predictors at each node, implemented as parameter "mtry". To increase robustness, we set the number of trees to 1000, which is twice the default value. The parameter "mtry" is optimized for maximum performance in the modelling process (and hardly sensitive). Additionally, we optimized model performance in a cross validation with five folds, as is commonly done (Murphy, 2012). However, unlike cross validation approaches in the classical sense, which randomly pick a set of individual data points, we divided the training data into five equal temporally contiguous chunks to avoid unrealistically good performance simply due to autocorrelation of temporally-close points in time. Finally in the 'reconstruction model', we derived annual suspended sediment yields by performing 250 Monte-Carlo realizations of the annual SSY for each year, which allows assessing the prediction uncertainty, from which the mean and quartiles of annual suspended sediment yields were computed. We chose the number of 250 Monte-Carlo realizations as it yields sufficiently robust estimates of the mean annual SSY (the confidence intervals of the mean are ca. ± 1.25 % of the mean) while keeping computation times reasonable.

### 3.3 Characteristics, sources and adjustments of input data

Here we describe the input data and their preparation for modelling. An overview of details such as coordinates of
stations and gauges, the temporal resolution of the different time series, their sources and data availability can be found in table A1 in the appendix.

#### 3.3.1    Discharge data, precipitation and temperature data

Gauge Vent has been operated continuously by the Hydrographic Service of Tyrol since 1967 (Strasser et al.,
2018) and discharge has been measured at 15 minute resolution through water stage recordings, complemented by a radar probe since 2000. At gauge Vernagt, discharge has been recorded since 1974 (Bergmann and Reinwarth, 1977) and is determined through water stage recordings and tracer-calibrated stage-discharge relations. Additionally, this is complemented by ultrasonic stage measurements, providing nearly uninterrupted discharge series since 1974 (Braun et al., 2007). The temporal resolutions of the discharge time series at both gauges varies
over time (from five and ten to 60 minutes, for details see table A1 in the appendix).

Precipitation and temperature have been recorded close to the gauge in Vent (see Fig. 1) since 1935 and are available at daily resolution. Both time series showed gaps, which creates a problem when using the chosen QRF approach. Thus, to fill the gaps in the precipitation time series, we used data recorded at gauge Vernagt: We derived a linear model between all days when daily precipitation sums were available from both stations and used the
resulting linear model for conversion (i.e. $P_{Vent} = P_{Vernagt}/1.3$). Likewise, temperature data recorded at Martin-Busch-Hütte (see Fig. 1) were aggregated to daily means and used to fill gaps in the temperature time series ($T_{Vent} = T_{MB} * 2.8089$). As a result, 2 % of the precipitation and 0.25 % of the temperature time series were filled (see Fig. A1 in the appendix).

At gauge Vernagt, precipitation and temperature have been recorded in high temporal resolution (5 to 60 minutes,
for details see table A1 in the appendix) next to the gauge since 1974. To fill the present gaps, we used data recorded by the Hydrographic Service since 2010 in close proximity to the Vernagt station. As some gaps still remained, we subsequently used data recorded at Martin-Busch-Hütte (conversion factors: $Temp_{Vernagt} = -0.002536 \cdot Temp_{MB}^2 + 0.9196*Temp_{MB} - 0.474$; $Precip_{Vernagt} = 0.895 \cdot Precip_{MB}$). With this, 12 % of the precipitation time series and 9 % of the temperature time series were filled, although many of these filled gaps occur during the
winter months, when the discontinued discharge data inhibit QRF modelling anyway (see Fig. A1 in the appendix). Some gaps still remain, but these, too, are mostly restricted to winters. We excluded the data from 1974 from the analyses at gauge Vernagt, as data were not available for the entire year.

#### 3.3.2    Turbidity and suspended sediment concentration data

At gauge Vent, turbidity has been measured since 2006 using two optical infrared turbidity sensors (Solitax ts-line and Solitax hs-line by Hach). To calibrate the turbidity measurements to suspended sediment concentrations (SSC), water samples are taken manually from the stream close to the turbidity sensors frequently (Lalk et al., 2014). Turbidity measurements are paused every winter (between October and April) to avoid damage to the equipment.

However, the equipment is reinstalled early enough to capture the spring rise in concentrations, and winter
sediment transport can be considered negligible (Schmidt et al., 2022b).

At gauge Vernagt, water is diverted into a measuring chamber (Bergmann and Reinwarth, 1977), where turbidity can be recorded while avoiding damage to the equipment by large rocks in the main channel. Turbidity was recorded in the summers of 2000 and 2001 (Staiger-Mohilo STAMOSENS 7745 UNIT) (Naeser, 2002) as well as 2019 and 2020 (Campbell OBS501). Water samples for calibration of turbidity to SSC were taken directly next to
the turbidity sensor, by hand in 2000 and 2001 (57 samples), and by means of an automatic sampler (ISCO 6712) in 2019 and 2020 (131 samples). The latter initiated sampling if one of two criteria was met: (i) regular sampling, to avoid long gaps between two samples, (ii) threshold-based sampling to obtain samples across the whole range of possible turbidity values. Gravimetric sediment concentrations $SSC_g$ were then determined in the laboratory and used to convert turbidity to SSC (2000-2001: SSC [g/l] = 0.1583 * turbidity [V] $^{-13.0877}$ (Naeser, 2002);
2019/20: SSC [g/l] = 0.00212*turbidity [FNU]).

### 3.3.3     Aggregation/disaggregation to different temporal resolutions

As temporal resolutions varied between the different time series, we aggregated or disaggregated the data to achieve homogenous temporal resolutions for the respective QRF models in daily, hourly and 10-minute
resolution. Data in 10-minute resolution were only needed for the gap-filling model at gauge Vernagt. For this, we had to disaggregate precipitation and temperature data from 60-minute resolution in 2000 and 2001, by dividing hourly precipitation sums by 6 and replicating mean hourly temperature values for the six corresponding 10 minute time steps.

For the analysis of annual Q, P and T, we summed up daily discharge volumes as derived from daily mean
discharge ($Q_{sum}$ [m³/day] = 60 * 60 * 24 * $Q_{mean}$ [m³/s]), added up daily precipitation sums and computed annual averages of daily mean temperature. At gauge Vernagt, we only considered data between May 1$^{st}$ and September 30$^{th}$ of each year due to inconsistent gaps in winter temperature and precipitation measurements.

### 3.4  Analysis of the results
### 3.4.1     Units, conversions and performance measures

From suspended sediment concentrations (SSC, modelled and from turbidity) and discharge (Q), we calculated sediment discharge $Q_{sed}$ [mass/time] (for analyses in high temporal resolution), daily Q-weighted SSC averages $SSC_{daily}$ [kg/m³] (i.e. the total daily sediment discharge $\Delta t \cdot \Sigma Q_{sed}$ divided by the daily discharge volume $\Delta t \cdot \Sigma Q$; this assigns more weight to time steps with higher discharge, which are more influential for sediment export), annual
suspended sediment yields SSY [t/a] and specific annual suspended sediment yields sSSY [t/a/km²] (for comparability among the gauges) as follows:

$$Q_{sed}(t) = SSC(t) \cdot Q(t),$$     (1)

where Q is discharge [m³/s],

$$SSC_{daily} = \frac{\Delta t \cdot \Sigma \, Q_{sed}(t)}{\Delta t \cdot \Sigma \, Q(t)},$$     (2)

where $\Delta t$ is the corresponding temporal resolution [s] and $\Sigma$ sums over all timesteps of the day,

$$SSY(year) = \Delta t \cdot \sum Q_{sed}(t) \text{ , and}$$

(3)

$$sSSY(year) = \frac{SSY(year)}{A},$$

(4)

where A [km²] is the catchment area and Σ sums over all timesteps of the year.

To quantify model performance and for our validation, we used the Nash-Sutcliffe efficiency index:

$$NSE = 1 - \frac{\sum_{i=1}^{n}[SSC_{obs}(i) - SSC_{mod}(i)]^2}{\sum_{i=1}^{n}[SSC_{obs}(i) - \overline{SSC_{obs}}]^2} ,$$

(5)

where 'obs' and 'mod' refer to observed and modelled SSC values and $\overline{SSC_{obs}}$ is the mean of observed SSC values (Zimmermann et al., 2012; based on Nash and Sutcliffe, 1970). The NSE is dimensionless and ranges from -∞ to 1, and a model with NSE over 0.5 can be considered accurate (Moriasi et al., 2007; Mather and Johnson, 2014). However, sediment export in the study area is highly seasonal (see also Schmidt et al., 2022) so that the NSE might be misleading, as models reproducing seasonality but fail to reproduce smaller fluctuations can still report a good NSE value (Schaefli and Gupta, 2007). Thus, we additionally computed the normalized benchmark efficiency as follows:

$$BE = 1 - \frac{\sum_{i=1}^{n}[SSC_{obs}(i) - SSC_{mod}(i)]^2}{\sum_{i=1}^{n}[SSC_{obs}(i) - SSC_{bench}(i)]^2} ,$$

(6)

where SSC_bench refers to the benchmark model suspended sediment concentration at timestep i. Commonly, this benchmark model is the mean of the observations for every Julian day over all years within n (i.e. the mean annual cycle) (see, e.g., Pilz et al., 2019). However, this is heavily influenced by individual events in our case, so we used the 60-day moving average of the mean SSC for every Julian day instead. As for the NSE, a BE value of 1 corresponds to perfect agreement of simulation to measurements and a model with BE > 0 is able to reproduce dynamics better than simply using statistics (Pilz et al., 2019).

### 3.4.2 Methods for trend analysis and change point detection

In order to quantify time series behavior, we generally followed the approach of first analyzing for existence of a trend. If a trend was detected, we assessed whether the trend was homogenous by analyzing for change points. If a change point was identified, we then examined the resulting segments of the time series for trends.

Most of the investigated time series are not normally distributed and some show autocorrelation. Thus, to calculate trend significance, we used the non-parametric Mann-Kendall test for linear trend detection in a version that was modified to detect trends in serially correlated time series (Yue and Wang, 2004) as recommended by (Madsen et al., 2014). To estimate trend magnitude, we used Sen's slope (SS) estimator (Sen, 1968). Both methods are implemented in the "mkTrend" function of the R-package "FUME" (Santander Meteorology Group, 2012). In our results, we only plot and refer to trends that were significant at least at a significance level α = 0.05 after correction for autocorrelation.

For change point (CP) detection, we used the non-parametric Pettitt's test (Pettitt, 1979), which is commonly used as it is a powerful rank-based test for a change in the median of a time series and robust to changes in distributional form (Yue et al., 2012), as implemented in the R package "trend" (Pohlert, 2020). However, it only gives one change point location without uncertainty quantification around its location and was shown to be sensitive to the

position of the change point within the time series, i.e. detection at the beginning and end of the series is unlikely (Mallakpour and Villarini, 2016).

Thus, as a complementary advanced approach that counterbalances the weaknesses of Pettitt's test, we used Bayesian regression with change points as implemented in the R-package "mcp" (which stands for "multiple change points"; hereafter we refer to this method as "MCP") (Lindeløv, 2020). This represents a much more flexible approach which allows to asses uncertainty through the resulting posterior distributions of the change point location and is applied in an increasing number of studies in different fields of research (e.g. Veh et al., 2022; Yadav et al., 2021; Pilla and Williamson, 2022). Although MCP allows to detect multiple change points, we only considered one change point as we aimed to detect the largest shift in the time series and to ensure comparability with Pettitt's test. Unless specified otherwise, we used the uninformative default prior, allowed free slope estimation before and after the change point, assumed a disjoined slope (i.e. step-like change) at the change point and allowed for changes in variance at the change point.

As mentioned earlier, we computed 250 Monte-Carlo realizations of the annual SSY as a result of the QRF model. We propagated this uncertainty by applying the trend and change point detection methods not only to the mean estimates but also to the 250 resulting time series realizations.

All calculations were done in R, version 4.2.1 (R Core Team, 2018).

## 4. Results Part I – Model evaluation

### 4.1. Validation A: Influence of reduction of temporal resolution

The temporal resolution for long-term reconstruction is limited to daily, as the respective long-term predictor data are available only at daily resolution at gauge Vent. As a result, we can expect some loss of information, e.g. on short-term precipitation intensity, which can be crucial for sediment dynamics. To assess the error magnitude, we ran two variants of the models at gauge Vernagt, in daily and hourly resolution based on all available training data (i.e. 2000, 2001, 2019 and 2020). We then compared daily sediment discharge ($Q_{sed}$) calculated from the out-of-bag model estimates to $Q_{sed}$ calculated from measured turbidity (Fig. 3a). It is important to stress that out-of-bag (OOB) estimates of any given day represent model predictions of only those regression trees where the particular day was not part of the training data. Both models reproduced daily $Q_{sed}$ very well, where the daily model showed a larger scatter, which is also reflected in the slightly lower NSE and BE. The comparison of annual sSSY (Fig. 3b) showed very similar estimates in most years. To rule out that model performances were strongly influenced by discharge (which is both a predictor in the model and used to calculate $Q_{sed}$), we also compared hourly and daily out-of-bag SSC instead of $Q_{sed}$. Yet the resulting NSE of 0.97 and 0.82 and BE of 0.95 and 0.73 for the hourly and daily models, respectively, still represent very good model performance in general and show that the loss of model skill between the two models because of different temporal resolutions is acceptable. Thus, we used the daily resolution models at both gauges in the following analyses for better comparability and applicability to the full length of the available time series.

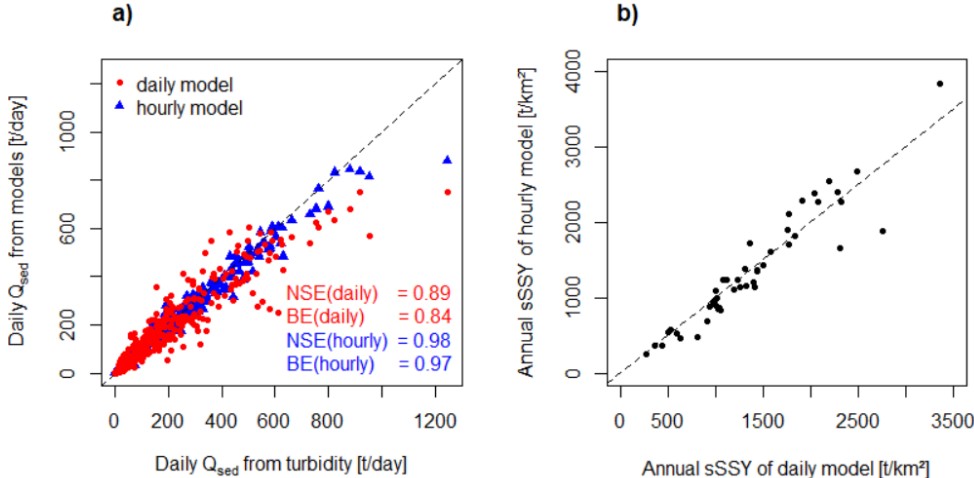

**Figure 3: a) Daily Q_sed calculated from out-of-bag prediction of daily and hourly models vs. Q_sed calculated from turbidity at gauge Vernagt; b) Comparison of mean annual sSSY estimates of the daily and hourly models at gauge Vernagt. NSE: Nash-Sutcliffe-efficiency; BE: benchmark efficiency. Dashed lines indicate ratio of 1:1.**

### 4.2. Validation B: Capability of temporal extrapolation

To evaluate the capability of the QRF models to reconstruct past sediment dynamics, we trained a daily model on the 2019-2020 (n = 212) data at gauge Vernagt and used the data of 2000-2001 (n = 367) for validation. The comparison of $Q_{sed}$ determined from turbidity to modelled $Q_{sed}$ showed that the model underestimates high daily

$Q_{sed}$ (Fig. 4 a). Nevertheless, the NSE of 0.73 and BE of 0.66 are indicative of a good representation of sediment dynamics in daily resolution. Comparing mean daily measured and modelled SSC instead of $Q_{sed}$, the NSE of 0.51 and BE of 0.33 still represent a satisfactory model performance (Moriasi et al., 2007; Pilz et al., 2019). Conversely, when training the model on the data from 2000 and 2001 and estimating $Q_{sed}$ (and SSC) for 2019 and 2020, performance was slightly lower, with an NSE of 0.6 (0.45) and BE of 0.38 (0.15) (Fig. 4 a).

In the 'validation model' results, annual yields were affected by overestimation in late August 2000 and underestimation in July 2001 (Fig. 4 b). As a result, the mean annual sSSY estimates were 31 % higher and 16 % lower than observed annual sSSY in 2000 and 2001, respectively. Considering the spread of the model results (i.e., the 2.5 and 97.5 percentiles of the 250 model predictions as depicted by the whiskers in Fig. 4 b, which were chosen as they are more robust than the extremes while covering 95 % of the estimates), 19 % overestimation

compared to the maximum model estimate in 2000 and 4 % underestimation compared to the minimum model estimate in 2001 remained. Unfortunately, annual yields of the validation model trained on the 2000-2001 data cannot be compared in the same way, since observations show gaps in both 2019 and 2020 (see also caption of Fig. 4 a).

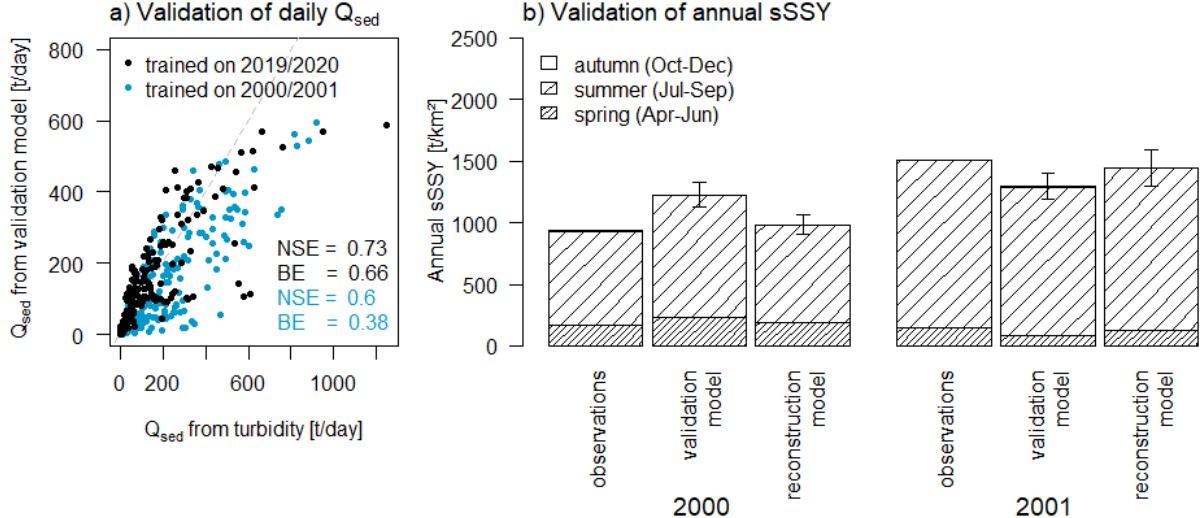

**Figure 4: a) Daily Q$_{sed}$ estimates from QRF validation models vs. Q$_{sed}$ from turbidity at gauge Vernagt. b) Annual sSSY based on turbidity observations, estimates of the validation model (without 2000-2001 data) and estimates of the reconstruction model (all training data) in 2000 (left) and 2001 (right). This comparison was not possible for 2019 and 2020, since observation data are missing in May - July 2019 and May - June 2020. Boxes depict mean model estimates and whiskers depict 2.5 and 97.5 percentiles of model predictions. NSE: Nash-Sutcliffe-efficiency; BE: benchmark efficiency.**

At gauge Vent, more years of data were available, so that the results of the five-fold cross-validation provided a more detailed picture of the temporal extrapolation ability (Table 1). Some cross-validation periods showed rather low NSE values, which indicates that they are harder to predict with only the remaining training data, and therefore contain more valuable data for training the full QRF model. For example, the period 2012 to 2014 with the lowest NSE contains the most extreme event in August 2014, which was likely linked to mass movements (Schmidt et al., 2022b). Similarly, the period 2018 to 2020 also shows a low NSE and a mass-movement event was observed in 2020 (ibid.). Including such periods in the training data apparently is of importance to the fidelity of the model. Conversely, the same cross-validation with a sediment rating-curve shows that QRF performance is much better, especially in those periods that are harder to predict. Consequently, QRF clearly excels over the traditional approach (see also section 4.6).

*Table 1 Results of the five-fold cross-validation at gauge Vent for QRF and sediment rating curves (SRC), with respect to mean daily SSC compared to turbidity measurements, expressed as Nash-Sutcliff efficiency (NSE) of the model estimates for each 1/5 of the time series in the cross validation (with corresponding years in brackets) and the full models (OOB = out-of-bag in the case of QRF).*

|  | NSE (OOB) full | NSE (2006 – 2008) | NSE (2009 – 2011) | NSE (2012 – 2014) | NSE (2015 – 2017) | NSE (2018 – 2020) |
|---|---|---|---|---|---|---|
| QRF | 0.6 | 0.48 | 0.55 | 0.21 | 0.69 | 0.39 |
| SRC | 0.41 | 0.41 | 0.33 | 0.09 | 0.61 | -0.055 |

### 4.3. Exceedances of predictor ranges in the past

At gauge Vent, maximum Temperature of the training period (T$_{train, max}$) was not exceeded during the reconstruction period. Maximum discharge of the training period (Q$_{train, max}$) was overstepped on 4 days in 1987 and precipitation

within the reconstruction period was larger than the maximum precipitation during the training period ($P_{train,\,max}$) on 3 days within the reconstruction period.

At gauge Vernagt, $Q_{train,\,max}$ was exceeded 6 times in the reconstruction period, and four of these days occurred in 2003. There was one day at gauge Vernagt, were $T_{train,\,max}$ was exceeded in 2017. $P_{train,\,max}$ was exceeded on 5 days within the reconstruction period, however on 2 of these days, the temperature was below zero and discharge was very low, so we considered these days negligible for annual sediment export.

There were also days at both gauges when discharge was lower than the minimum discharge measured discharge during the training period ($Q_{train,\,min}$). Yet all of these days were in April, May or October, when SSC are very low, and as very low discharge also translates to very low transport capacities, we considered the error negligible for annual sediment yields.

### 4.4. Performance of the reconstruction models

To test the performance of the reconstruction models, we compared out-of-bag SSC estimates of the reconstruction models and SSC estimates of sediment rating curves (trained on all available data) to turbidity measurements at both gauges (Fig. 5).

QRF performance is superior to sediment rating curves (SRC) at both gauges with respect to mean daily SSC: as can be seen in Fig. 5 and the NSE and BE values, SRC are generally less capable of capturing the variability in the training data. The NSE and BE values of the QRF model represent a satisfactory performance at gauge Vent and very good performance at gauge Vernagt (Moriasi et al., 2007). Considering mean annual SSC at gauge Vent, QRF performs much better than SRC. At gauge Vernagt, QRF performs very well and slightly better than SRC, but there are only 4 years available for comparison, which confines the representativity of this analysis.

Performance with respect to $Q_{sed}$ is slightly better for both models as compared to SSC. SRC estimates yield values for NSE of 0.73 and BE of 0.61 at gauge Vernagt and NSE of 0.49, and BE of 0.35 at gauge Vent. Still, QRF performance is superior (NSE of 0.89 and BE of 0.84 at gauge Vernagt and NSE of 0.64 and BE of 0.54 at gauge Vent). For both models, performance is better at gauge Vernagt than at gauge Vent.

Rare days with extremely high SSC (and $Q_{sed}$ above ca. 7000 t/day) are systematically underestimated by both models. However, it is important to emphasize that Fig. 5 shows out-of-bag (OOB) predictions, i.e. the QRF model predictions for these "extreme" days if the particular day in question is not part of the training data. To quantify the effect of this underestimation of daily $Q_{sed}$ on annual SSY for the QRF-model, we calculated the difference between measured and estimated daily $Q_{sed}$ (OOB) for the 10 days with the highest $Q_{sed}$ in the turbidity time series at both gauges. The differences correspond to 0.6 to 2.8 % of the annual SSY at gauge Vernagt, and 1.7 to 19.1 % of the annual SSY at gauge Vent. However, the 19.1 % underestimation are attributable to the most extreme event in the time series, where 26 % of the annual SSY were exported within 25 h in August 2014, likely in association with a mass movement event (see Schmidt et al., 2022). The full model estimate of the 'reconstruction model' (i.e. not OOB estimates) for this day shows an underestimation of only 6 %.

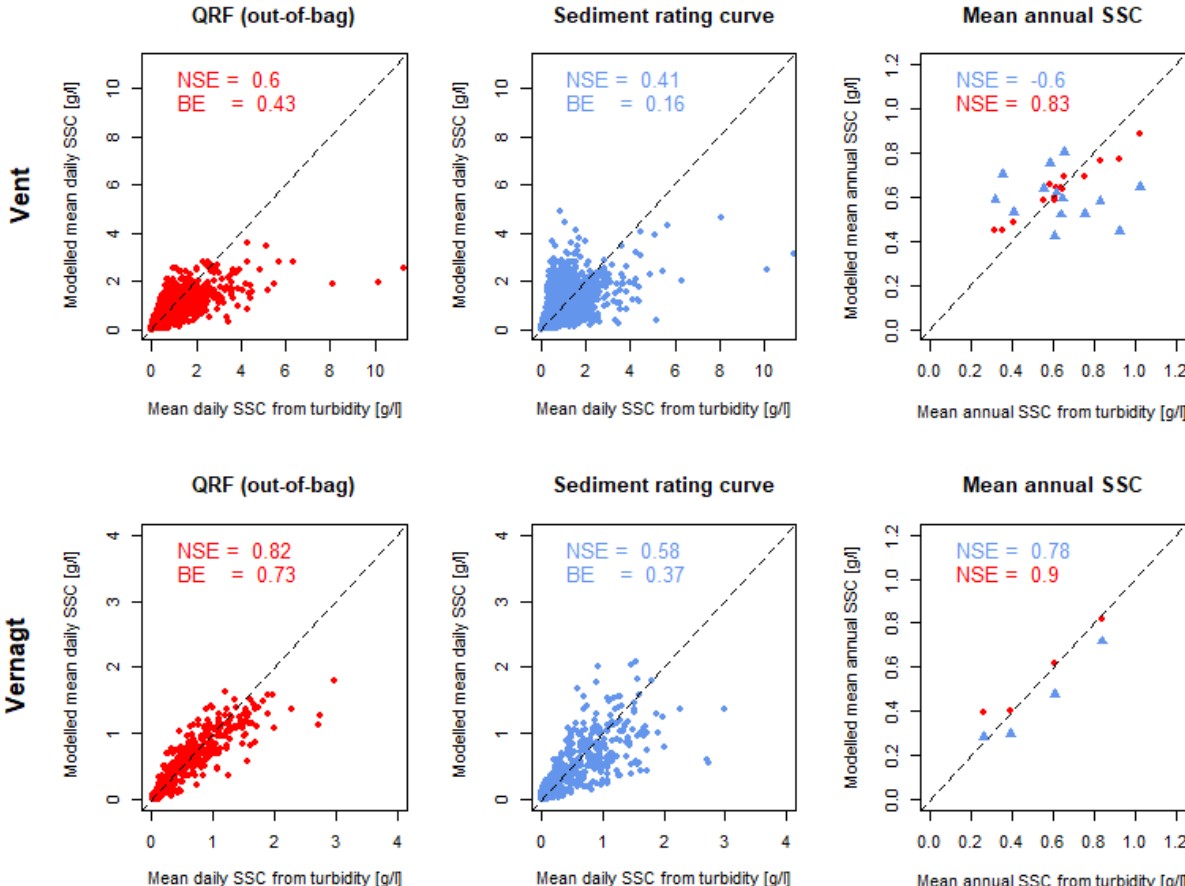

Figure 5: Measured versus modelled sediment concentration and yield using QRF (OOB-estimates, red) and SRC (blue). Top row: gauge Vent; bottom row: gauge Vernagt; left: QRF, daily values, centre: SRC, daily values, right: QRF and SRC, annual values. NSE: Nash-Sutcliffe-efficiency; BE: benchmark efficiency.

## 5.    Results – Part II: Analysis of estimated annual sSSY, predictors and mass balances

### 5.1.    Annual sSSY and their development over time

In the resulting time series, the average sSSY of all years (± 1 standard deviation) is 1401 (± 453) t/km²/yr at gauge Vent and 1383 (± 668) t/km²/yr at gauge Vernagt. This indicates overall similar magnitudes of sediment export per catchment area, yet with much higher variability at gauge Vernagt. To assess how suspended sediment dynamics changed over time, we analyzed the time series of annual sSSY at the two gauges for trends and change points.

At both gauges, mean modelled annual sSSY show significant positive trends (Fig. 6). In Vent, 97.6 % of the 250 time series realizations show significant positive trends (Sen's Slope (SS): 9 – 15 t/km²/a) and at gauge Vernagt, all realizations show significant positive trends (SS: 28 – 35 t/km²/a). At gauge Vent, Pettitt's test yields a significant change point in 1981 in annual sSSY, which is also true for 99.6 % of the realizations (for 1 realization, the change point was detected in 1980). Accordingly, the MCP analysis shows a rather narrow probability density distribution around 1980/81 for all realizations with maximum probability density in 1981.

At gauge Vernagt, MCP shows very similar probability density distributions to that of Vent with maximum probabilities in 1981 for all realizations. However, Pettitt's test detects a change point in 1989 (p < 0.01; 66 % of realizations in 1989 and 27 % in 2002). As we elaborate in the discussion, this is likely due to a limitation of the Pettitt's test. Thus, we divided both time series in 1981 to examine the resulting segments for trends.

In the first segment, no significant trends were detected at gauge Vernagt, and at gauge Vent, only two of the 250 realizations show significant positive trends (SS of 11.1 and 3.9 t/km²/a; see Fig. 5). In the second segment (i.e. after 1981), we detected a negative trend in mean annual sSSY at gauge Vent (SS = - 7.6 t/km²/a), as well as in 42 realizations (SS of - 13.1 to - 5.9 t/km²/a). In contrast, at gauge Vernagt mean sSSY (SS = 23.5 t/km²/a) as well as 248 of the realizations (SS of 20.2 to 27.9 t/km²/a) show strong positive trends. The average sSSY (± 1SD) of all years after 1981 is 1579 (± 391) at gauge Vent and 1537 (± 603) at gauge Vernagt.

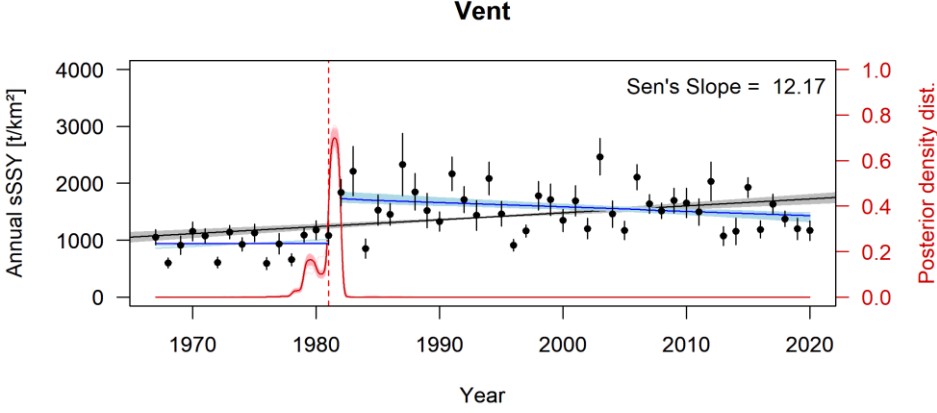

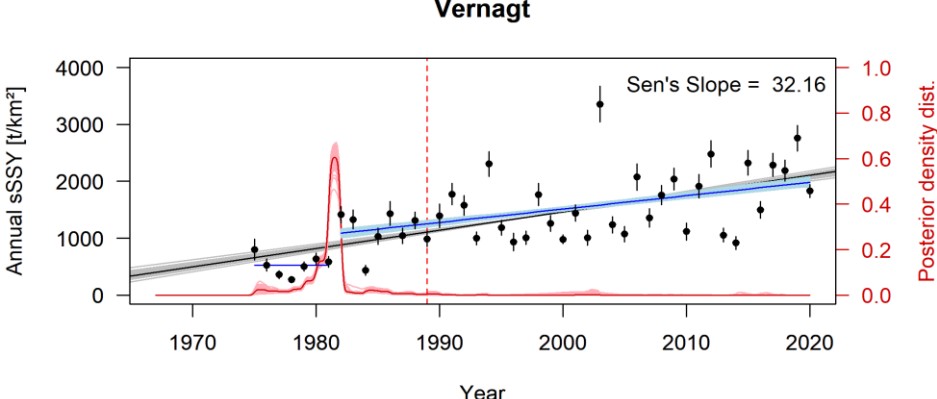

**Figure 6: Mean specific annual suspended sediment yields (sSSY) as reconstructed by the QRF model (black points). Whiskers depict 2.5 % and 97.5 % quantiles of the 250 QRF realizations. Overall trend of mean annual sSSY is given by Sen's Slope (black line) and trends of all 250 QRF realizations in grey. Change points determined by Pettitt's test (red dashed line) and MCP (posterior distributions of 250 realizations as solid light red and mean as dark red lines). Trends in the time series segments before and after 1981 are given as dark blue (mean) and lightblue (realizations) lines (only plotted if significant).**

### 5.2. Results – Part III: Analysis of predictors and glacier mass balances

We found positive trends in annual sSSY at both gauges with high probabilities of a step-like increase around 1981. To assess whether this coincides with changes in temperature (i.e. changes in snow and / or glacier melt and discharge) or changes in precipitation, we analyzed time series of the predictors, temperature, discharge and precipitation, as well as glacier mass balance data with respect to trends and change points.

Annual specific discharge sums at both gauges show significant positive overall trends (Fig. 7 a) and b). In Vent, both change point detection methods indicate high probabilities of a change point in 1981. At gauge Vernagt, annual discharge volumes roughly doubled within the examined period, from ca. 1250 mm to ca. 2500 mm. The

two change point (CP) detection methods disagree as in the case of annual sSSY, as the Pettitt test detects a change point in 1990 while MCP suggests a change point with high probability in 1981. Dividing both time series in the year 1981, both segments of the Vent time series show insignificant trends, while both segments at gauge Vernagt show significant positive trends (1st segment: Sen's slope = 32.7 mm/a, p < 0.01; 2nd segment: Sen's slope = 17.2 mm/a, p < 0.001).

We analyzed mean monthly discharge volumes and found significant positive trends in May and June in Vent (a detailed figure can be found in the Appendix, Fig. A2). As with temperature, change points are indicated by both methods in 1981 for July discharge, and around 1995 for June discharge. At gauge Vernagt, discharge shows significant positive trends from May through August (see figure A3 in the Appendix). In June, a change point (CP) is indicated around 1995 by both methods, and in July, MCP indicates a CP around 1981, but the Pettitt's test does not yield a significant change point. For May and August discharge, Pettitt's test yields change points in the late 1990s and mid-1980s, respectively, but MCP shows very widespread probability distributions.

Summer precipitation sums (May – September) at gauge Vent (Fig. 7 c) show no significant trend, while annual precipitation sums show a significant positive trend. A change point is identified by the Pettitt's test in 1991 (p < 0.001) while MCP yields a widespread probability distribution with the highest probability in the late 1990s. At gauge Vernagt, we confined the analysis to summer precipitation (May – September) as the time series is affected by gaps in some winters (see Fig. A1 in the appendix). Summer precipitation at gauge Vernagt shows a significant positive trend (Fig. 7 d). Pettitt's test yields a significant change point in 1992 while MCP gives a very widespread distribution.

We derived mean annual temperatures at gauge Vent, however, at gauge Vernagt, we computed mean summer temperatures (between May and September) instead, as temperatures are missing in winter in many years (see also Fig. A1 in the appendix). Both timeseries show significant positive trends but no clear change points, as Pettitt and MCP disagree and MCP yields very widespread probability distribution (Fig. 7 e) and f). Additionally, we analyzed mean monthly temperatures over time and found that while most months show positive trends, at both gauges July is the only month with a high change point probability around 1981 (Fig. 7 e) and f). At both stations, July temperatures in 1982 and 1983 were exceptionally warm.

In addition to the hydro-climatic predictors considered in the QRF model, we analyzed independent data of annual mass balances of the Vernagtferner (VF) and Hintereisferner (HEF), the two largest glaciers in the Vent catchment with long glaciological mass balance records. Since 1965 (VF) and 1952 (HEF), the two glaciers have been regularly and extensively surveyed for volume changes (World Glacier Monitoring Service, 2021; Strasser et al., 2018). Annual mass balances of both glaciers show significant negative trends (fig. 7 g) and h). Significant change points are indicated in 1985 for both glaciers by the Pettitt test, while MCP attributed (much) higher probabilities for change points to the year 1981. Notably, annual mass balances of both glaciers are almost exclusively negative after 1980 (with the exception of 1984, where mass balances are barely positive). Dividing the time series in 1981, no significant trends are detected in the first segments of both time series and significant negative trends in the second segments (HEF: -20 mm/a; VF: -17 mm/a).

Summer mass balances were only (continuously) available for the Vernagtferner and show a strong negative trend and both Pettitt and MCP detect change points in 1981 (Fig. 7 h). No significant trends are detected in the resulting time series segments. Winter mass balances do not show a significant trend nor change points.

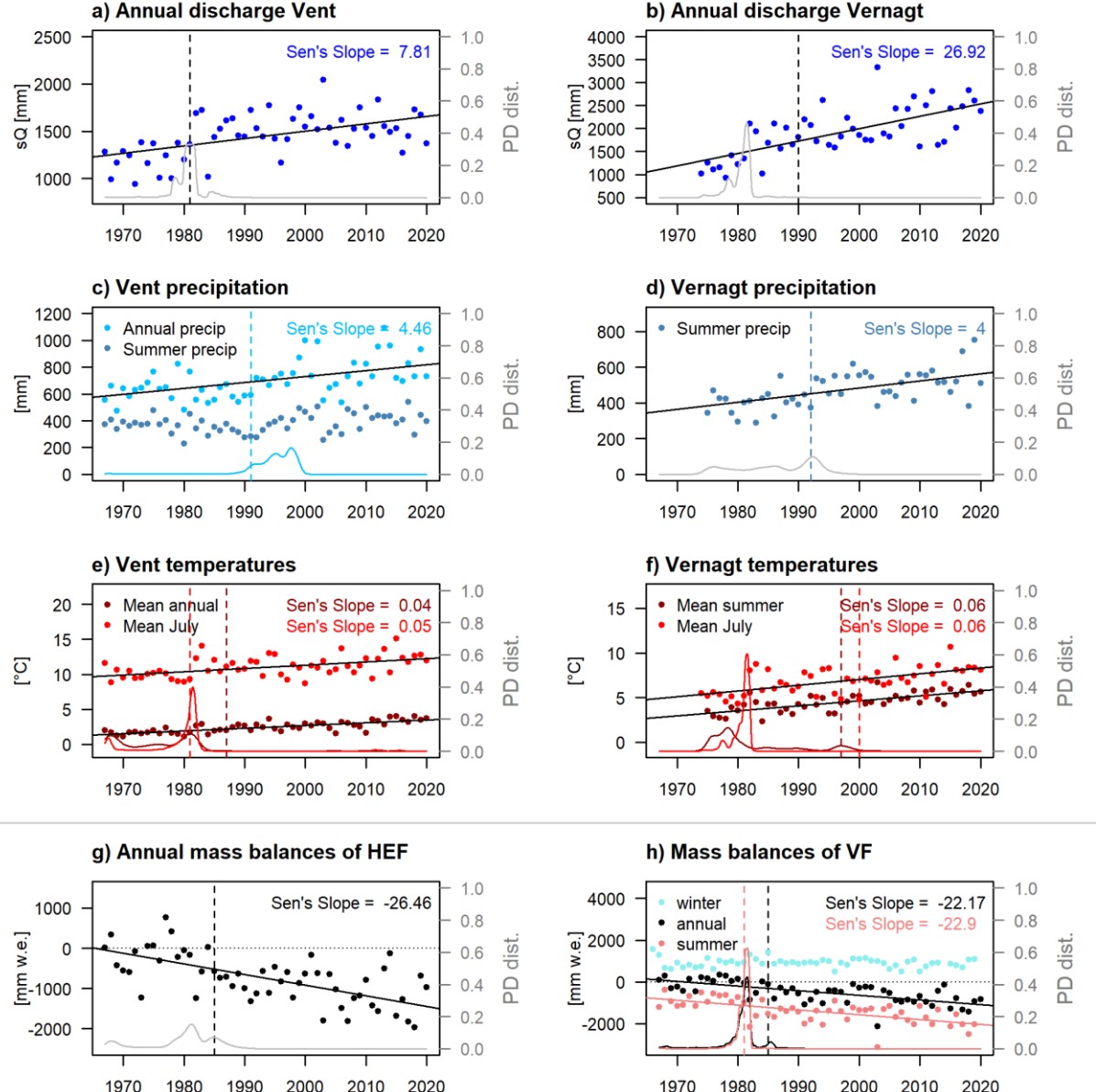

**Figure 7: a) and b) annual discharge yields at gauges Vent and Vernagt; c) Annual and summer (May – Sept) precipitation in Vent; d) Summer (May – September) precipitation at gauge Vernagt; e) Mean annual and July temperatures in Vent; f) Mean summer (May – Sept) and July temperatures at gauge Vernagt; g) Annual mass balances of the Hintereisferner (HEF); h) annual, winter and summer mass balances of Vernagtferner (VF). Dashed vertical lines show change point locations according to Pettitt's test and solid lines show Sen's Slope. Both are only drawn if they are significant at α = 0.01. Lines at the bottom show posterior density distributions of MCP change point locations, with colors corresponding to the respective variables, if several variables are depicted within one plot.**

## 6. Discussion

### 6.1. Model evaluation

The presented study aimed to examine extensively, whether Quantile Regression Forest is a suitable method for estimating past sediment export. Overall, the final QRF reconstruction models outperformed sediment rating curves (SRC) by about 20 percent-point in explained variance at both gauges – even though we assessed out-of-bag (OOB) estimates of QRF versus rating curves based on all available data. OOB performance was better at gauge Vernagt (NSE of 0.82 with respect to daily SSC) than at gauge Vent (NSE of 0.6), although even the latter

corresponds to a satisfactory performance (Moriasi et al., 2007). We suggest that there are three reasons for this difference in performance. First, the Vent catchment is much larger (almost 100 km²) than the 11.4 km² Vernagt catchment. Thus, suspended sediment dynamics at gauge Vent integrate over more different processes, which adds to the complexity compared to the mostly melt-driven Vernagt catchment. Second, precipitation measurements at gauge Vent are unlikely to be representative of the entire catchment, due to the catchment size and especially due to topographical effects. Thus, more localized rainstorms are likely to be captured poorer. And third, SSC in Vent reach much higher values than at gauge Vernagt, so that uncertainties in the turbidity measurement are more relevant: at low concentrations, the light emitted by the turbidity sensor is predominantly scattered by solid particles, while at high concentrations, absorption becomes the dominant process. This causes the relation between the light detected by the photo sensor in the turbidity probe and SSC to become non-linear (Merten et al., 2014) and leads to a much higher variance in the SSC-turbidity relationship. However, despite these effects and the lower NSE with respect to daily $Q_{sed}$ at gauge Vent, annual sSSY show very good agreement. Thus, we conclude that – given the availability of a large enough and varied training dataset and sufficiently long records of the predictors – QRF represents a reliable tool with the ability to broaden our understanding of the response of high-alpine areas to climate change in the past decades.

Nevertheless, a few limitations need to be considered. As a major limitation, QRF cannot extrapolate if predictors in the reconstruction period exceed the range of values represented in the training dataset (Francke et al., 2008a). In these cases, the modelled values will potentially suffer from over- or underestimation. At the same time, assessing the number of such exceedances in the period of model application provides an indication of the representativity of the training data set, which must be sufficiently large and varied for such data-driven approaches (Vercruysse et al., 2017). In our case, exceedances were rare (seven days at gauge Vent and 10 days at gauge Vernagt), so we consider the error negligible for annual sediment export. We strongly encourage future studies to assess the number and severity of such exceedances for this purpose, especially in view of changing conditions, e.g. due to (prospective) climate change.

More generally, the QRF model can only reproduce the quantitative and qualitative conditions as represented within the training data. This has several implications. First, the spread of the QRF model results needs to be interpreted as a minimum estimate of uncertainty, as it can only reproduce the "known unknown", i.e. the variability represented in the training data and their respective relationships or the lack thereof. And second, major qualitative changes in the functioning of the modelled system, such as changes in connectivity e.g. through the formation of proglacial lakes or large-scale storage of sediments along the flow paths, cannot be captured. However, for the catchments considered in this study, such major changes are at least not indicated in historic aerial images (Laser- und Luftbildatlas Tirol, 2022) and the longitudinal profile of the major water courses are very steep, which precludes significant sediment storage.

We found that our models at both gauges underestimated rare, high daily SSC and $Q_{sed}$ (figures 3 – 5). This is not surprising given that these events are rare and that figures 3 to 5 show out-of-bag predictions, which means that the respective estimates are based only on those trees that have seen few or none of these conditions. Including them in the training of the final reconstruction model alleviated this effect.

Further, we suspected that the aggregation of precipitation and discharge to daily values might involve some loss of information e.g. on sub-daily precipitation intensity and maximum discharge, which would very likely affect sediment export estimates. Indeed, the underestimation is a little more pronounced in the daily model as compared to hourly resolution, yet the difference was relatively small (Validation A, Fig. 3 a). Adding to this, the

underestimation at the daily scale does not seem to propagate to annual estimates, as high annual sSSY are not systematically underestimated to the same extent (Fig. 5b) and the underestimation of rare events at the daily scale has a limited effect on the annual estimates (section 4.4). This is in accordance with the finding that only about ¼ of the annual yield in Vent is transported during (precipitation) events (Schmidt et al., 2022b), as opposed to other fluvial systems where the majority of the annual sediment yield is transported by several extreme events (Delaney et al., 2018). Consequently, the proposed QRF model is also applicable at daily resolution, enabling its application to the longer time series available.

We assessed temporal extrapolation ability at gauge Vernagt by training a model on the data of 2019/20 and comparing the estimates to measurements in 2000 and 2001 (Validation B; Fig. 4), which showed over- and underestimation of annual sSSY by 31 % and 16 %, respectively. In interpreting these results, it has to be considered that the amount of data used for training is very small and less than half of the final reconstruction model (212 of 579 days), while the amount of training data is known to be crucial in data-driven models (e.g. Vercruysse et al., 2017). Specifically, turbidity recordings only started in mid-July 2019, so that only one spring season was available for training. Thus, given the rigorousness of this test as well as the temporal distance of 20 years, we find that the validation yields satisfactory results with good agreement of dynamics on short timescales as well as annual estimates. The validation model trained on the data of 2000/2001 yielded lower but still satisfactory NSE values. We attribute this to higher discharge and temperature values in 2019 and 2020, which, again, points at the importance of exceedances. With better data availability at gauge Vent, we assessed temporal extrapolation ability through a five-fold cross-validation. This showed that some periods were harder to model based solely on the remaining data. This means that these periods contain rather distinct data and are thus especially valuable as training data for the full QRF model, as they contained the highest SSC and $Q_{sed}$ in the time series. We repeated this cross-validation with the sediment rating curves (SRC). It proved to be inferior to QRF in most periods, which showed that in our case QRF is better able to extrapolate from limited data, especially with respect to periods containing extreme events, which are difficult to describe per se, due to threshold effects such as the activation of mass movements (Zhang et al., 2022). This seems to contradict the fact that, numerically, a SRC is indeed capable of extrapolating beyond the range of the training data, whereas QRF is not. As QRF still performed better in modelling periods with extreme events in the cross-validation, we attribute this to the circumstance that QRF is able to model interactions and is not bound to a linear or monotonous relationship between the predictors and SSC estimates. Apparently, these features are more important and influential than extrapolation in the sense of sheer "extension of a curve".

### 6.2. Analysis of annual specific suspended sediment yields (sSSY), predictors and mass balances

The overall magnitudes of annual yields at the two analyzed gauges fall at the high end compared to an extensive collection from the European Alps (Hinderer et al., 2013) and are in good accordance with yields from other catchments in the Stubai and Ötztal Alps (Schöber and Hofer, 2018; Tschada and Hofer, 1990) (see also Schmidt et al., 2022). Their reconstruction for the past five decades constitutes an important contribution to the understanding of long-term sediment budget, as other existing records commonly only cover some decades.

The reconstructed annual sSSY at both gauges show overall positive trends and change points around 1981. Indeed, the two change point detection methods did not agree at gauge Vernagt, where MCP yielded high change point probabilities around 1981, whereas Pettitt's test detected a significant change point in 1989 (or 2002 in some realizations). Similarly, the results did not agree for July temperatures and discharge at gauge Vernagt. We attribute

this to a limitation of Pettitt's test, which is known to be much less sensitive to break points located near the beginning or end of the time series (Mallakpour and Villarini, 2016). Thus, we conclude that there is a high probability of change points around 1981 in the reconstructed sSSY as well as discharge and July temperature time series at gauge Vernagt, as the MCP probability distributions are very narrow and similar to the corresponding probability distributions in Vent.

Our results suggest that the step-like increase in modelled sSSY is linked to the onset of increased ice melt. Mean annual temperature in Vent and mean summer temperature at gauge Vernagt at both locations show gradual positive trends without a clear change point. Yet, mean July temperatures show high change point probabilities around 1981 at both locations, which is probably heavily influenced by extraordinarily high temperatures in July 1982 and 1983. July temperatures are especially relevant for firn and glacier melt, since July is the month with the highest firn and ice melt contribution to discharge, after snow melt contributions have peaked in June and snow cover has decreased substantially (Kormann et al., 2016; Weber and Prasch, 2016; Schmieder et al., 2018; Schmidt et al., 2022b). This shift is reflected in discharge, which shows a step-like increase around 1981 at both gauges and continues to be elevated after this change point. The analyses of monthly discharge showed that July was the only month with a change point around 1981 at both gauges, which again emphasizes the dominance of increased firn and ice melt. In contrast, we did not find evidence for a change in precipitation sums around 1981, which would indicate that enhanced hillslope erosion on snow- or ice-free surfaces played a crucial role in the sSSY increase. With regard to suspended sediment dynamics, it is conceivable that the increase in ice melt translates to an increase in sediment-rich glacial meltwater (Delaney and Adhikari, 2020) as well as intensified fluvial erosion of sediment-rich proglacial areas.

These changes in the predictors of our QRF model are also reflected in the mass balance time series of the Hintereisferner and Vernagtferner, which were almost exclusively negative after 1980, with very negative summer mass balances in 1982 and 1983. Escher-Vetter (2007) and Abermann et al. (2009) have also shown that mass loss started in 1981 at both glaciers, resulting from negative summer balances. With respect to the Vernagtferner, the drastically higher ablation area ratio of almost 80 % in 1982 as compared to around 25 % in the preceding years (Escher-Vetter and Siebers, 2007) indicates that large areas of the glacier became snow-free in 1982. This entails crucial changes in albedo and therefore intensified ice melt and thinning of firn areas due to rising energy absorption at the glacier surface (Escher-Vetter, 2007; Braun et al., 2007). We interpret this as a regime shift as summer mass balances continue to be lower than before 1981 although (July) temperatures decrease again, with the exception of 1984, which was characterized by a relatively high number snowfall days during the ablation period (Escher-Vetter and Siebers, 2007).

Similar step-like increases in sediment export have been reported from the upper Rhône basin in Switzerland (Costa et al., 2018) (only slightly later, around 1985, which is likely due to lower average temperatures), as well as for the headwaters of the Yangtze River on the Tibetan plateau (Li et al., 2020; Zhang et al., 2021) and 28 headwater basins in High Mountain Asia (Li et al., 2021). Thus, we deem it unlikely that the change points identified in the present study are related e.g. to a change in measurements, also because they were detected at both gauges in the same year and in both (July) temperature and discharge. More generally, substantial increases in sediment export in response to climate change have been reported from cold environments around the globe (e.g. Bogen, 2008; Koppes et al., 2009; Bendixen et al., 2017b; Singh et al., 2020; Vergara et al., 2022) and are in accordance with state-of-the-art conceptual models, which expect that phases of glacier retreat (and re-advance)

lead to the highest increase in sediment yield across glacial cycles (Antoniazza and Lane, 2021), as has been confirmed by several studies (Lane et al., 2017, 2019; Micheletti and Lane, 2016).

Interestingly, we found opposing trends in the time after 1981, with strongly increasing annual sSSY at gauge Vernagt and decreasing sSSY at gauge Vent. To some extent, this is reflected in changes in discharge, where we found no significant trend (and Sen's slope close to zero) after 1981 at gauge Vent, but a strong positive trend at gauge Vernagt. Altogether, this could indicate different timings of "peak sediment" (Ballantyne, 2002; Antoniazza and Lane, 2021) and thus a stabilization or compensation of the larger Vent catchment as opposed to the nested Vernagt catchment. Some independent observations support this nothion: For example, the glacier tongue of the Hochjochferner (denoted as "HJF" in Fig. 1) , located in the southernmost tributary valley to the Rofental, has retreated by about 2 km since the 1970s (determined based on historic aerial image collection of the State of Tyrol (Laser- und Luftbildatlas Tirol, 2022)) and has now retreated behind a rock sill. Additionally, several small lakes have formed in its glacier foreland that likely act as sediment sinks. Another two smaller glaciers in the North-Eastern part of the Vent catchment, Mitterkarferner and Platteiferner ("MKF" and "PF" in Fig. 1), have disappeared almost completely since the 1970s. Conversely, the Vernagtferner glacier has also experienced considerable loss in area and volume, but lacks such pronounced qualitative changes thus far.

## 7. Conclusions

In the presented study, we tested a Quantile Regression Forest (QRF) model, which enabled the estimation of sediment export of the past five decades for the two gauges Vent and Vernagt. This allowed to analyze annual specific suspended sediment yields (sSSY) for trends and change points. Annual sSSY show positive trends as well as step-like increases after 1981 at both gauges. As this coincides with exceptionally high July temperatures in 1982 and 1983, distinct changes in the glacier mass balances of the two largest glaciers in the catchment area and a sudden increase in ablation area of one of the glaciers, we conclude that temperature-driven enhanced glacier melt is responsible for the step-like increase in sSSY. This is also mirrored in discharge measurements at both gauges, which show change points around 1981 as well. Opposing trends after 1981 at the two gauges could indicate different timings of 'peak sediment'. These analyses also demonstrated the value of assessing change points in addition to trend analyses, in order to detect sudden changes in the analyzed geomorphic systems and thus facilitate a better understanding of critical time periods.

Further, we explored advantages and limitations of the QRF approach. As a major limitation, QRF may yield underestimates if predictors exceed the range of values represented in the training dataset. To estimate the effect on the model results, we suggest to assess the number such exceedances, which can also be leveraged to evaluate the representativity of the training dataset. Further, we found that events with very high SSC and $Q_{sed}$ tend to be underestimated. While this is not surprising given the rareness of such events, the effect of such underestimations on the annual yields was small. The assessment of the temporal extrapolation ability revealed a satisfactory performance of QRF and illustrated again the importance of a sufficiently varied dataset, i.e. including larger events. A comparison of QRF to sediment rating curves in a five-fold cross-validation showed that QRF was better able to model periods that contain very high SSC caused by mass movements. This points to the favorable ability of QRF to model threshold effects, which is a major advantage compared to approaches bound to continuous relationships. The final QRF reconstruction models showed good performance at both gauges and outperformed SRC by about 20 percent-points of explained variance. We conclude that the presented approach is a helpful tool for estimating past sediment dynamics in catchments where long enough SSC measurement series are lacking.

Future studies could help gain more knowledge on decadal-scale sediment export from high alpine areas by applying the presented approach to other catchments. In turn, advancing knowledge on past changes will support and prepare the development of management and adaptation strategies.

**Code availability**

The code of the Quantile Regression Forest model including the preprocessed data and raw model results is available on B2Share (Schmidt et al., 2022a).

**Data availability**

See column 'data availability' in table A1 in Appendix A.


**Author contribution**

LKS developed the general idea and conceptualized the study with TF and PG, with mentoring and reviewing of AB. LKS gathered the raw data. LKS, PG and TF installed and maintained the automatic water sampler, for gauge Vernagt heavily supported from the installations run by CM, who also supplied past data. PG prepared the input

data, adapted and extended the model code and performed modelling experiments with support and supervision of TF and LKS. LKS conducted the statistical analyses with support by TF. CM reviewed and evaluated the results in the glaciological context. LKS prepared the original draft including all figures, and all authors contributed to writing of this paper.

**Acknowledgements**

This research has been supported by the Deutsche Forschungsgemeinschaft Research Training Group "Natural Hazards and Risks in a Changing World" (NatRiskChange GRK 2043/1 and GRK 2043/2 grants) as well as a field work fellowship of the German Hydrological Society (DHG).

We acknowledge the support of the Hydrographic Service of Tyrol, Austria, for providing data as well as logistical

support and fruitful discussions.

We thank Oliver Korup for his encouraging advice on statistical analyses, Matthias Siebers, Andreas Bauer, Irene Hahn, Theresa Hofmann, Marvin Teschner, Nina Lena Neumann and Joseph Pscherer for their help and support during field and laboratory work as well as Stefan Achleitner, Carolina Kinzel and the Environmental Engineering laboratory at the University of Innsbruck for their kind support during field and laboratory work. We thank three

anonymous reviewers for their thorough and valuable comments that helped enhance and refine this manuscript.

**Competing interests**

The contact author has declared that none of the authors has any competing interests.

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

Appendix A

**Table A2: Characteristics of input data. Q = discharge, T = temperature, P = precipitation, SSC = suspended sediment concentration. HD = Hydrographic Service of the State of Tyrol, Austria. BAdW = Bavarian Academy of Sciences and Humanities, Munich, Germany.**


| Variable | Location | Coordinates (lon, lat; WGS84) | Length | Temporal resolution | Source | Availability |
|---|---|---|---|---|---|---|
| **Q** | Vent | 10 54 39, 46 51 25 | 01.01.1967 – 31.12.2017 | Daily means | HD | (eHYD, 2021) |
| | | | 01.01.1967 – 31.12.2020 | 15-minute mean | | Can be requested via wasserwirtschaft@tirol.gv.at |
| **T, P** | Vent | 10 54 46, 46 51 26 | 01.01.1935 – 31.12.2011 | Daily mean (T) and sum (P) | Institute of Meteorology and Geophysics, Innsbruck, Austria | (Institute of Meteorology and Geophysics, 2013) |
| | | | 01.01.2012 – 31.12.2016 | Daily mean (T) and sum (P) | Institute of Atmospheric and Cryospheric Sciences, Innsbruck, Austria | (Juen and Kaser, 2017) |
| | | | 01.01.2017 – 31.12.2020 | Daily mean (T) and sum (P) | HD | Can be requested via wasserwirtschaft@tirol.gv.at |
| **SSC (turbidity)** | Vent | 10 54 39, 46 51 25 | 01.05.2006 – 31.10.2020 | 15-minute mean | HD | (Schmidt and Hydrographic Service of Tyrol, Austria, 2021) |
| **Q, P, T** | Vernagt | 10 49 43, 46 51 24 | 01.05.1974 – 31.10.2001 | 60-minute mean | BAdW | (Escher-Vetter et al., 2012) |
| **Q** | Vernagt | | 01.05.2000 – 20.10.2001 | 10-minute mean | | |
| **Q, P, T** | Vernagt | | 01.05.2002 – 31.12.2012 | 5-minute mean | | (Escher-Vetter et al., 2014) |
| **Q, P, T** | Vernagt | | 01.05.2013 – 15.10.2020 | 5-minute mean | | Data will successively be made available on PANGEA. |
| **SSC (turbidity)** | Vernagt | | 01.05.2000 – 20.10.2001 | 10-minute mean | | |
| **SSC (turbidity)** | Vernagt | | 30.04.2019 – 03.11.2020 | 5-minute mean | | |
| **SSC samples** | Vernagt | | 23.05.2019 – 30.08.2020 | 131 samples | This study | |
| **T, P** | Vernagt (HD) | 10 49 42, 46 51 24 | 07.10.2010 – 31.12.2020 | 15 minute mean (T) and sum (P) | HD | Can be requested via wasserwirtschaft@tirol.gv.at |
| **T, P** | Martin-Busch-Hütte | 10 53 18 46 48 03 | 23.09.2010 – 31.12.2020 | 15-minute mean (T) and sum (P) | HD | Can be requested via wasserwirtschaft@tirol.gv.at |

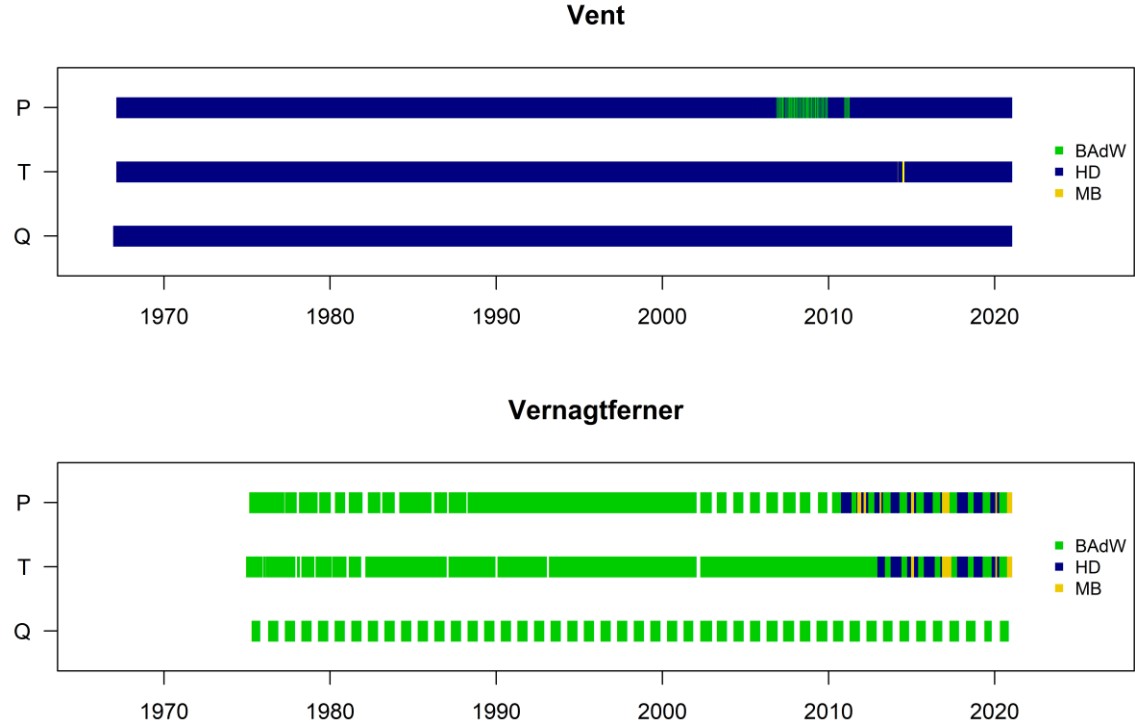

**Figure A1  Discharge (Q), Temperature (T) and Precipitation (P) data at the two gauges after gap-filling procedure. The color indicates the data source (BAdW = Bavarian Academy of Sciences and Humanities, HD = Hydrographic Service of Tyrol stations at gauges Vent (top) and Vernagt (bottom), MB = Martin-Busch Hütte).**

**Vent**

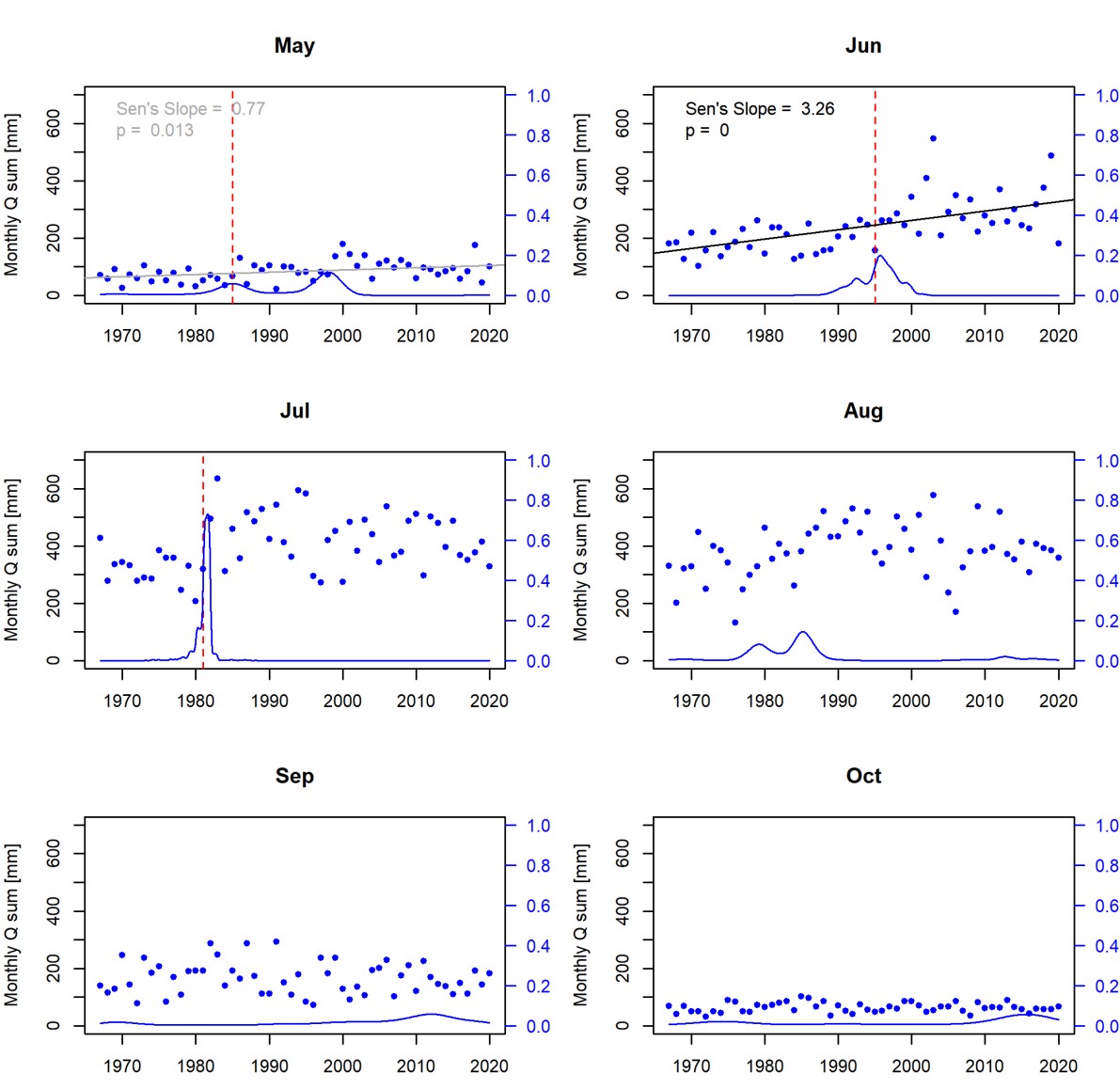

*Figure A2 Trends and change points in monthly discharge sums at gauge Vent. Dashed red lines indicate change points according to Pettitt's test (significant at α = 0.01), blue lines represent change point probability distributions of MCP, solid black (and grey) lines indicate trends according to Mann-Kendall test significant at α = 0.01 (and 0.05).*


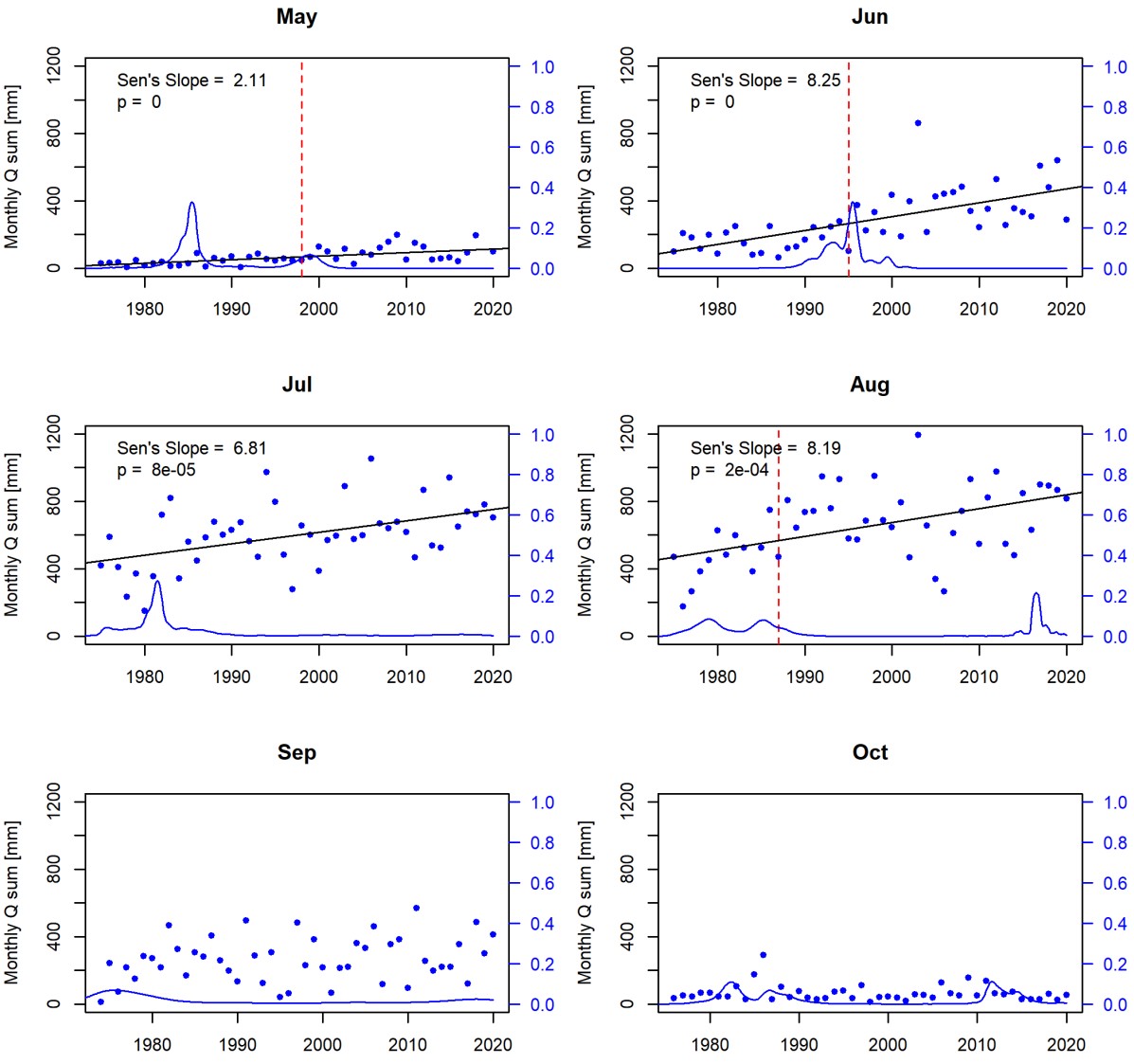

*Figure A3 Trends and change points in monthly discharge sums at gauge Vernagt. Dashed red lines indicate change points according to Pettitt's test (significant at α = 0.01), blue lines represent change point probability distributions of MCP, solid black lines indicate trends according to Mann-Kendall test significant at α = 0.01..*
