# Peer review of "Reconstructing five decades of sediment export from two glaciated high-alpine catchments in Tyrol, Austria, using nonparametric regression"

_EGUsphere, 2022_

## Author Comment (AC2)

[Figure]

*Figure 1 Comparison of sediment rating curve to QRF performance when both are trained solely on daily data from 2000/01 (left) and on all available training data (2000, 2001, 2019, 2020; right).*

[Figure]

*Figure 2 Performance of QRF (a) and sediment rating curves (b) compared to mean daily SSC derived from turbidity measurements at gauge Vent. Panel c) shows mean annual SSC estimates based on QRF (red circles) and SRC (blue triangles).*

*Table 1 Results of cross validation at gauge Vent with respect to mean daily SSC given as nash-sutcliffe efficiency (NSE) of out-of-bag predictions of the full model (Nash_OOB_full), for each 1/5 of the time series in the cross validation ("nash1" to "nash5") and the mean NSE of the 5 cross validation periods.*

| nash_OOB_full | nash1 | nash2 | nash3 | nash4 | nash5 | meanNS |
|---|---|---|---|---|---|---|
| 0.61 | 0.48 | 0.55 | 0.21 | 0.69 | 0.39 | 0.46 |

---

## Author Comment (AC3)

[Figure]

*Figure 1 Mean annual SSC estimates over time with change points determined by Pettitt test (dashed) and with the mcp package (solid line).*

[Figure]

*Figure 2 Left: Variable importance at gauge Vent. Right: Variable importance at gauge Vernagt*

---

## Author Response (AR1)

Dear Roberto Greco, dear anonymous referees,

We would like to thank you again for your very detailed comments, questions and suggestions. Below, we provide our response as direct answers to each comment and point out the respective changes to the manuscript. Please be aware, that the line numbers and chapters mentioned in the "Changes made" sections refer to the latest version of the manuscript.

We hope that our changes will be to your satisfaction.

Best, Lena Katharina Schmidt on behalf of all authors

**10 RC1: 'Comment on egusphere-2022-616', Anonymous Referee #1, 04 Oct 2022**

**# General comments**

5

In this manuscript, the authors applied machine learning to reconstruct sediment discharge records in two catchments in the Austrian Alps. After validating the reconstructed record, the

15 authors identified trends and regime shifts with various change point detection methods. They identify the early 1980s as a turning point for the sediment dynamics and suggest links with temperature-driven glacier dynamics.

This is a valuable contribution showcasing the application of modern, data-driven methods to

- 20 a field where they are yet to be routinely applied. However, beyond its technical value, the paper falls short from connecting its methods and results to the wider literature and addressing how such methods could be applied to other areas of study. For example, the discussion section would benefit from circling back to the larger scope and scientific questions mentioned in the introduction.
- 25 **Changes made:** We have restructured and re-written the discussion section to better refer to the larger scope, as well as the conclusions.

Overall, the paper is well-structured easy to follow, but key information is missing from the Methods section for readers both familiar and unfamiliar with the techniques applied (see specific comments below).

30 *#* Specific comments

**Inconsistent verb tenses**

In Methods and Results section, verb tenses switch between past and present. Some authors prefer to use present all along, while some prefer to use past to describe all past actions including methods and results. This is the authors' choice, but it has to be consistent. For

example, L152, the authors use "we train" to describe past training, then L157 the authors use "we applied" to describe past application. This is inconsistent and is found in a number of places.

Answer: *Thank you. We will harmonize the use of tenses.* Changes made: *We have harmonized the use of tenses throughout the manuscript.*

40 *##* Differences in precipitation gradients

The authors mentioned L126 that the precipitation gradient is 0.05 per 100. At L175, the correction factor between P(Vent) and P(VF) is P(Vent) = 1/1.3 \* P(VF) = 0.769 \* P(VF). Using the elevation from gauges at Vent (1891 m) and Vernagt (2635) leads to an elevation difference of 744 m. The correction factor calculated from the previously cited precipitation

45 gradient is then 744 / 100 \* 0.05 = 0.372 and equals roughly half of the reported value. I understand that the authors used the recorded data to derive their value, but I am curious for the large difference between the value reported and the one cited.

Answer: Thank you for this interesting question. Schöber et al. (2014) state 4-5 % per 100 m for the area, but that includes a neighbouring valley (around Obergurgl) as well.

50 However, Vent receives considerably less precipitation than Obergurgl, due to its shielded location between the highest mountain in Tyrol (Wildspitze 3770m) and Ramolkogl (3550) and because it is located further away from the alpine ridge (luv/lee effects). This may be why the difference in measurement time series is larger than expected from the gradient.

**Any ensemble of models can assess model uncertainty**

- 55 L230-232: I disagree with that statement. The quantification of the uncertainties that the authors attribute to QRF is a result from ensembles of model with a random component. One could get a distribution of predicted values from an ensemble of neural networks with random initialization, or random partitions between training and testing. Ensemble of neural networks is not uncommon: in deep learning literature, results for new neural networks are often
- 60 reported from a 10-fold cross-validation for which 10 models are trained, and, sometimes, the ensemble of these 10 models used for predictions. I would suggest the authors clarify the advantage of QRF if I misunderstood it, or be more nuanced in this statement and back it to QRF ensemble process rather than to QRF itself.
- Answer: Thank you for this comment. It seems we have to be more clear about the QRF
  approach, which inherently includes ensemble processes (to produce a "forest" of regression trees). If we understand it correctly, this is not inherent to the other methods you mentioned. We suggest to improve the description in this segment and add "traditional" (i.e. "compared to traditional fuzzy logic or ANN").

**Changes made:** We have improved the description (L 235 et seqq.).

70 ## Key information missing when describing QRF, too much information for change point detection

Key information is missing when describing QRF:

- L320: The authors mention here that the time series used as predictors show autocorrelation. Is there also some correlation between the time series? If so, this could be leveraged by

- 75 methods like ARIMA or NARX to perform the predictions. In general, it is not best practice for machine learning approaches to only use one approach, and tree-based approach are not often the go-to algorithm(s) to perform time series predictions. I recommend that the authors better justify their choice of using only one algorithm, and specifically QRF. This may be done summarizing the cited literature, but is at the moment insufficient by itself.
- 80 Answer: Thank you for these suggestions. It seems we have to express more clearly that the scope of the study was to test QRF specifically in the alpine catchments (as it had been applied to sediment dynamics successfully in the past) and interpret the results rather than identifying the best possible method in a comparison. Although there might be other applicable methods, we find that QRF works sufficiently well with the presented data.
- 85 To our knowledge, there are no studies directly comparing QRF to other approaches for sediment concentration modelling – except the one we already mentioned: Compared to other methods, that are traditionally applied for suspended sediment concentration modelling, QRF performance was superior (Francke et al., 2008). As reviewer 2 suggested to compare QRF to sediment rating curves – a very simple and traditional approach for estimating sediment
- 90 concentrations we will add that to compare QRF with it. However, a study comparing random forest (which QRF is based on) to support-vector machines and artificial neural networks for suspended sediment concentration modelling (Al-Mukhtar, 2019) concluded that performance of random forest was superior. A study on the prediction of lake water levels (i.e. not with respect to sediment concentrations, but at least

hydrological timeseries) came to the same conclusion (Li et al., 2016). 95 We suggest to improve the description of the aim of the study. **Changes made:** We have improved the description of the aim of the study (L. 88 et seqq.).

- L243: The authors mention here that they used a 5-fold cross validation. While crossvalidation is often performed with 5 or 10 folds, it is also common practice to perform

repeated cross-validation to have more robust statistics on model performance. It would be 100 beneficial if the authors justified the number of folds (i.e. why 5 instead of 10), and the choice of not doing any repeats.

Answer: Thank you for this detailed question. We will point out more clearly that - unlike "usual" cross validations - we use temporally contiguous blocks of our data for the cross-

- validation, to avoid unrealistically good performance simply though autocorrelation. This 105 would be an issue if we just allowed to pick individual days for the cross-validation. Thus, ours is a rather strict approach and repeats in the classical sense are not as easily possible. Beyond that, the number of folds is indeed always arbitrary to some extent. We tried to find a compromise between too selective test data and too few training data. Choosing 5-fold cross
- validation as a compromise roughly corresponds to the number of complete seasons included 110 in the shortest time series at VF.

**Changes made:** We have improved the description in the manuscript (L. 260 et seqq.).

- L325-339: The level of details provided here for change point detection departs from the

115 level of details provided in the section detailing QRF. In particular, the QRF section does not mention any implementation details. I deem these details to be unessential. In particular, the names of the R packages are unnecessary.

**Answer:** We do not fully agree here, since the stating of the R packages, which in our view is common practice, promotes reproducibility and acknowledges the work of others. With the

- 120 respect to the implementation details of QRF, we build upon other publications and published the code alongside the manuscript, which we hope facilitates reproducibility. **Changes made:** We have provided more details on QRF, by adding a description and explanation of the used predictors and improving the description of the optimization and the ancillary predictors (i.e. antecedent conditions) (section 3.2.). We also added a reference to
- the github repository, where the model version used in former studies can be found (section 125 3.2, L 237).

Nonetheless, the term "mcp" is used throughout the paper but never defined; please provide a clear definition of it and use an uppercase acronym instead of the package name.

Answer: Thank you, we will do that. 130 **Changes made:** *We defined the term "mcp" in line 386 et seqq.*

Beyond the justification of using the Mann-Kendall tests, there is a lack of references justifying the use of these specific change point detection methods, and a reader with a

different perspective may ask why the authors did not use another method (for example, the 135 Fisher Information; https://doi.org/10.3390/w14162555 for a recent example in hydrologic sciences).

**Answer:** Thank you for this suggestion. Indeed, there are many available change point detection methods. We intended to apply an established, often-applied method (Pettitt, e.g. by

Costa et al., 2018) and – in contrast to most studies, that only use one method - counter-140 balance its weaknesses (no uncertainty quantification, low detection probability if change point is located near the beginning or end of the time series) by using another approach with complementary advantages, i.e. mcp, which is being applied in an increasing number of studies and research fields (e.g. (Veh et al., 2022; Yadav et al., 2021; Pilla and Williamson,

145 2022)). We will improve the description to make this decision more easily understandable to the readers.
Changes made: We have improved the description in lines 380 et seqq.

Furthermore, the choice of hyper-parameters for the QRF is crucially missing and should be
reported. It seems that the authors have not performed any tuning of the hyper-parameters which should also be justified.

Answer: The two most important hyper-parameters are the number of trees in a "forest" and the number of selected predictors at each node ("mtry" parameter). The latter is optimized in the modelling process (and is hardly sensitive). A larger number of trees increases robustness

155 (i.e. reduces the effect of the heuristic nature of QRF) – at the expense of computation time. We set the number of trees to 1000, which is twice the default value, to ensure robustness. We will add this to the description.

**Changes made:** We have added this to the description in L 256 et seqq.

- 160 ## Limits to applicability and links to introduction context and questions L551-559: In this paragraph, the authors could start discussing implications of the applicability of their method. For example, how lucky were the authors in finding such limited out-of-domain observations during the period for which they wanted to apply their model? Was that expected? Is that expected in the future if extreme conditions are more likely
- 165 (e.g. increased temperature, increased precipitation)? How does this impact the applicability of the same approach in other catchments, or over different timescales? In particular, could this be used at all for forecasting future evolution of sediment dynamics? All of these questions are interesting, and I suggest that the authors address at least a few of them to explain to the wider audience the limits of their approach. Specifically, this could be
- 170 mentioned in the Outlook section 6.4 to circle back to the wider themes of the introduction. Answer: Thank you for this interesting question. We do not think that the number of out-ofdomain observations is a question of "luck". Naturally, for data-driven approaches, datasets must be "sufficiently large"- and the larger and more varied the training dataset, the less likely occurrences of out-of-domain observations will be. Thus, this rather gives some
- 175 indication on the representativity of the training data and therefore also the credibility and limits of the model results. However, we agree that we should emphasize the need to assess this for future studies on other catchments and / or future evolution. Changes made: We have emphasized the possibility of using the number of exceedances to

changes made. We have emphasized the possibility of using the number of exceedances to evaluate the representativity of the training data set (conclusions, discussions (l. 629 et seqq.), and mentioned it in the abstract) and encouraged future studies to do so, especially with respect to future estimates.

**Minor specific comments**

185

- L245: "250 Monte-Carlo realizations": at this point in the manuscript, it is unclear on which random variable the Monte-Carlo simulation is performed. It became clear to me at L340, but the authors should probably add some clarification before that point. The number of Monte-Carlo simulation should also be justified. Why 250 iterations were chosen? If the authors used a convergence criterion, it should be reported and justified.

a convergence criterion, it should be reported and justified.
 Answer: We will improve the description in L245. Generally, a higher number of iterations will results in a more robust estimate of the mean annual suspended sediment yield. In practice however, this is one of the main points that will increase computation time. The chosen number of 250 iterations yields sufficiently good results. This can e.g. be seen in the

195 *confidence intervals of the mean estimates, that are*  $ca \pm 1.25$  *% of the mean.*

Changes made: We have made clear that we refer to annual SSY and added a justification of the 250 realizations in lines 266 et seqq.

L280: Is there a reason for choosing the partition of the data between data from 2019-2020
 for training and data from 2020-2021 for validation. Why not the other combination too (2020-2021 for training, 2019-2020 for validation)?

Answer: There seems to be a misunderstanding, it is not 2020/21 but 2000/01. Since we wanted to assess how well the model can reproduce past suspended sediment yields and dynamics, this seemed more relevant than using past data to reconstruct years that are more

- 205 recent. Moreover, this choice results in a stricter evaluation, because there are less training data available from 2019/20 than from 2000/01.
  If we train (and tune) the QRF model based on the 2000/01 data (hereafter QRF2000/01) and validate it against 2019/20, we find that QRF2000/01 performance is similar to QRF2019/20 with respect to SSC and not as good as QRF2019/20 with respect to SSY (see figure 1 below).
- 210 QRF2000/01 performance with respect to SSC is clearly better than SRC, performance. Changes made: We added the NSE and BE values for a model trained on the 2000 and 2001 data and validated against 2019 and 2020 in line 432 et seqq. and added the respective data points to figure 4 a).

215 Figure 1 Validation of QRF models and sediment rating curves trained on 2000 and 2001 data against 2019/20 data. Top: QRF; Bottom: Sediment rating curve; Left: SSC estimates; right: SSY estimates.

- L373: Why these percentiles were chosen?

Answer: We chose these percentiles because they are more robust than the extremes (i.e. min and max), and because they cover 95 % of all estimates, which is common in our perception.
 Changes made: We added a explanation in line 438 et seqq.

- L385-401: This 4.3 section seems like it should be mentioned in the Methods. I would suggest to place appropriate mentions of this in the Methods section, before such an important validation check on the methods is reported as a result.

225 Answer: We agree. We will move the first paragraph to the methods.

Changes made: We have moved the first paragraph to the methods (line 206 et seqq.).

- L575: "independently": I question the independence that the authors refer to here. One catchment is nested within the other, and the data at one location was used to correct the data

at the other location. This introduces some level of dependence between the two datasets thus they cannot be described as independent.
 Answer: Thank you. What we tried to express here, is that we deem it unlikely that e.g.

changes in measurements could have caused these shifts at both locations at the same time. The two gauges are nested, but the annual discharge at gauge Vernagt is only about 15 % of

- 235 the annual discharge in Vent, so if the increase had only occurred at gauge Vernagt, it would not necessarily be visible at gauge Vent, much less to this extent. Also, we need to clarify that only precipitation data at gauge Vent were corrected using precipitation data from gauge Vernagt. Discharge data and temperature time series were measured and used completely independently.
- 240 We agree that "independently" is not be the right word here and will correct that, yet we do not think this changes out conclusions.

Changes made: We improved the description in the methods, so that it becomes clear that only precipitation data at gauge Vent were corrected using data from gauge Vernagt and replaced the word "independently" by a more adequate description in lines 728 et seqq.

245

**Technical corrections**

- L57: Please clarify for who the timescales are relevant; relevant for management?

250 Answer: Thank you, we will clarify that we are referring to relevant timescales for investigating changes associated with anthropogenic climate change. Changes made: We have clarified as suggested (L50).

- L75: remove e.g.

- 255 Answer: There are more factors and we only named the most relevant ones for our case, which is why the e.g. makes sense here. More information can then be found in the cited paper (Huss et al., 2017).
   Changes made: We have left the e.g., as suggested.
- 260 L78: long enough data -> long term data
   Answer: Thank you, we will change this.
   Changes made: We have adjusted this, as suggested (L72).

L96: machine-learning -> machine learning; this term is never defined which would be
 beneficial for reader unfamiliar with it
 Answer: Thank you, we will add a definition.
 Changes made: We have added a short definition and a reference for further reading (L89).

- L97: In past studies: QRF has not only been used in geomorphology. I would suggest adding
a qualifier here to narrow the scope of the sentence
Answer: *Thank you, we will do that.*

```
Changes made: We have adjusted this, as suggested (L91 et seqq.).
```

- L102: data situations -> data availability
- 275 Answer: Thank you, we will change this.
   L103: bear -> leads to

Answer: Thank you, we will change this.

- L103: and taken together [...] -> so that, taken together, they give [...]

Answer: Thank you, we will change this.

280 - L104: location -> catchment

Answer: Thank you, we will change this. Changes made: We have adjusted these issues, as suggested (L106 et seqq.), although some have become obsolete because we rewrote the sentence.

285

- L106: with respect to trends, which -> for trends, some of which
   Answer: Thank you, we will change this.
   Changes made: We have adjusted this, as suggested (L109).

   L145: The legend for Figure 1 refers to gauge then catchment for the two areas of interest; it would be clearer if only one type was mentioned
   Answer: We attempted to describe it in the hydrologically correct way, thus we suggest leaving it as it is.
   Changes made: We have left this, as suggested (L148).
- 295 L173: in daily resolution -> at a daily resolution
   Answer: Thank you, we will change this.
   Changes made: We have adjusted this, as suggested (L287).
  - L190-191: I would move "since 2006" after "turbidity has been measured"
- 300 Answer: Thank you, we will change this. Changes made: We have adjusted this, as suggested (L305).

- L255: "developments": I am unsure what the authors mean here by developments: is it related to methods or evolution?

305 Answer: We are referring to long-term changes in catchment dynamics. We will clarify this. Changes made: We have adjusted this, as suggested (L176).

- L260: remove "truly" Answer: Thank you, we will do that. Changes made: We have removed this, as suggested (L181).

310 Changes made: We have removed this, as suggested (L181).

L267: extraordinary -> rare
 Answer: Thank you, we will change this.
 Changes made: We have adjusted this, as suggested (L188).

315

- L269: benefit of the opportunities -> benefit from these opportunities Answer: *Thank you, we will change this.* Changes made: *We have adjusted this, as suggested (L190).*

- L272: "fig. 2": the way figure are referenced is inconsistent: it is sometimes "fig", "Fig", or "figure". Please harmonize.
   Answer: Thank you, we will do that.
   Changes made: We have harmonized this throughout the manuscript.
- L279: repaired -> corrected; to match the language used in Fig. 2
   Answer: Thank you, we will adjust this.
   Changes made: We have adjusted this, as suggested (L201).

L280: 2000/01 -> 2000-2001; and everywhere else where the authors use this notation
 instead of the full years separated by an hyphen
 Answer: *Thank you, we will change this.* Changes made: We have adjusted this throughout the manuscript.

- L288: 3.2 Analysis of results: this section number is wrong as the previous section was already 3.3

Answer: Thank you, we will correct this. Changes made: We have corrected this.

- L291: [t/time]: use either dimension [mass/time] or units [t/day] not both; also consider replacing t by Mg

Answer: Thank you, we will change this to mass/time. Changes made: We have adjusted this, as suggested (L337).

L302: When introducing the Nash-Sutcliffe efficiency, it would be beneficial if the authors provide its range and directionality so that readers unfamiliar can interpret the following figures more easily by knowing that a value of one relates to good performance Answer: *Thank you, we will add this.* Changes made: *We have adjusted this, as suggested (L354 et seqq.).*

- L349: remove "As described earlier"
 Answer: Thank you, we will remove this.
 Changes made: We have removed this, as suggested.

- L350: in daily resolution -> at that resolution

355 Answer: Thank you, we will change this. Changes made: We have adjusted this, as suggested (L405).

- L350-351: rewrite this sentence; right now it reads as if the loss is crucial whereas it is the information or the impact of its loss that is

- 360 Answer: Thank you, we will change this. Changes made: We have adjusted this, as suggested (L405 et seqq.).
  - L386: please add a reference to this statement since "it is known" Answer: Thank you, we will add a reference. Changes made: We have added a reference (L208).
- **Changes made:** *We have added a reference (L208).*

- L418: A square exponent is missing in the units of the specific suspended sediment yield Answer: Thank you, we will correct this. Changes made: We have corrected this, as suggested (L516 et seqq.).

370

335

340

- L425-429: Should this two-sentence paragraph be merged with the previous paragraph? Answer: *Thank you, we combine this paragraph with the following paragraph.*. Changes made: *We have combined the paragraphs, as suggested.*

375 - L468: where -> for which, remove "which was"
 Answer: Thank you, we will change this.
 Changes made: This became obsolete, because we rewrote the paragraph to describe the newly added figure in the Appendix.

- L472: remove "in the time"; not significant -> no significant
   Answer: Thank you, we will change this.
   Changes made: We have rewritten this sentence (L531 et seqq.).
  - L506: before we discuss -> then we discuss
- 385 Answer: Thank you, we will change this. Changes made: We have decided to erase this paragraph in the course of restructuring the discussion.
- L511: the term "critical point" has very precise meaning in the study of dynamical system, I would advise using "significant change point" rather than "critical point".
   Answer: Thank you, we will adjust this.
   Changes made: This sentence became obsolete during the rewriting and restructuring of the discussion, but we do not use the term "critical point" in the manuscript.
- 395 L518: extraordinary -> rare
   Answer: Thank you, we will change this.
   Changes made: We have adjusted this, as suggested.
- L540: several reasons -> three reasons
  400 Answer: Thank you, we will change this.
- Answer: Thank you, we will change this.
   L541: Firstly -> First, L542: Secondly -> Second, L544: And thirdly -> Third
   Answer: Thank you, we will change this.
   Changes made: We have adjusted this, as suggested in these two comments (L612 et seqq.).
- 405 L550: please add a reference to this statement since "it is known"
   Answer: Thank you, we will add a reference.
   Changes made: We have added a reference (L627).
  - L641: gap of knowledge -> knowledge gap
- 410 Answer: Thank you, we will change this. Changes made: This sentence became obsolete during the rewriting and restructuring of the conclusions.

**415 RC2: 'Comment on egusphere-2022-616', Anonymous Referee #2, 18 Nov 2022**

I appreciate the opportunity to review the manuscript, entitled 'Reconstructing five decades of sediment export from two glaciated high-alpine catchments in Tyrol, Austria, using nonparametric regression'. The topic is study is of great importance to not only the earth and environmental science community but also the policymakers and

- 420 practitioners such as hydropower companies and water resource managers. This study presents an attempt to reconstruct the long-term suspended sediment export in alpine glacierized basins based on the available shorter records and machine learning. Despite some limitations, the proposed method is capable of reconstructing the sediment yield over the past decades with satisfactory performance.
- 425 **Major comment 1:** Based on modelling scheme in Figure 2, the model validation should target SSC, which is very reasonable and necessary. While, in the results section, the authors

only validate the performance of sediment discharge and sediment yield, which are the product of discharge and SSC. In your model (Quantile Regression Forest), discharge is also one of the model input variables and important predictors. The high validation coefficients

430 (NSE and BE) could be only part of the story and maybe just because discharge appears in both input and output variables. Thus, I would kindly suggest the authors try to re-validate the model performance using SSC and replace both Qsed and sSSY in Figure 3-5 with SSC as shown in figure 2 if possible.

Answer: Thank you for this comment. Indeed, we need to state more clearly, that e.g. the
 tuning of the models is performed on daily/hourly SSC (not daily Qsed). However, the
 quantity that we are ultimately interested in is (annual) sediment yield, as we want to
 understand whether the amount of sediment transported from the catchments changed over
 time. Adding to this, we find that yields are a more meaningful way to aggregate to annual
 resolution than mean annual SSC, because of the skewed nature of the concentration

- 440 distribution. In mean annual SSC, low concentrations on days at the beginning and end of the season are given the same weight as high concentrations during the glacier melt season when discharge is also high so actually, most of the sediment export happens during the glacier melt period. We believe that this can be captured better using sediment discharge and annual yields.
- 445 Thus, we suggest to add NSE and BE calculated on SSC to the text. As you can see below, the values do not change substantially, if we use SSC instead of Qsed in validation A (hourly vs. daily model resolution at gauge Vernagt, figure 3a): Hourly model: NSE(Osed) = 0.98 NSE(SSC) = 0.97

|     | nourry moder. | BE(Qsed) = 0.96,
BE(Qsed) = 0.97,  | BE(SSC) = 0.95                   |
|-----|---------------|---------------------------------------|----------------------------------|
| 450 | Daily model:  | NSE(Qsed) = 0.89,
BE(Qsed) = 0.84, | NSE(SSC) = 0.82 $BE(SSC) = 0.73$ |

In validation B (model trained on 2019/20 and validated against 2000/01 at gauge Vernagt), the NSE = 0.51 and BE = 0.33 still represent a satisfactory model performance (Moriasi et al., 2007; Pilz et al., 2019), as does model performance at gauge Vent (comparing SSC from

455 *turbidity to out-of-bag model estimates) with* NSE = 0.6 *and* BE = 0.43. *For mean annual SSC at gauge Vent, the NSE is even as high as for annual yields (*NSE(SSC) = 0.825 *vs.* NSE(SSY) = 0.832).

Changes made: We have stated more clearly, why we focus on SSY instead of SSC (L162) and added the NSE and BE based on SSC to the text (L413 et seqq. and L431 et seqq.).

- In the introduction, the authors say that "Quantile regression forests (QRF) (Meinshausen, 2006) are a multivariate non-parametric regression technique based on random forests, that have performed favorably to sediment rating curves" (paragraph 95). Although it is proven in other publications, I think this statement still needs to be tested and evaluated in this study. If possible, I would suggest the authors compare the SSC simulations by QRF model and SSC simulations by sediment rating curves and explicitly demonstrate how much improvement can
  - be done by the QRF model than sediment rating curves.

Answer: Thank you for this valuable comment. When comparing daily SSC estimates using sediment rating curves (SRC) to QRF at gauge Vernagt (VF), we find that SRC estimates are in fact slightly better in validation B, i.e. when we train both QRF and SRC solely on SSC from 2010/20 at gauge Vernagt and compare modelled to measured SSC values in 2000/01

470 from 2019/20 at gauge Vernagt and compare modelled to measured SSC values in 2000/01 (see figure 2 below). However, when using the full dataset, SRC performance is worse than QRF performance, even though QRF performance considers out-of-bag estimates only. Thus,

SRC performance gets worse with a larger training dataset, which already demonstrates that SRC cannot describe the variability in SSC as well as QRF.